# Effects of Water and Nitrogen Regulation on Apple Tree Growth, Yield, Quality, and Their Water and Nitrogen Utilization Efficiency

**DOI:** 10.3390/plants13172404

**Published:** 2024-08-28

**Authors:** Xingqiang Li, Siqi Li, Xiaolin Qiang, Zhao Yu, Zhaojun Sun, Rong Wang, Jun He, Lei Han, Qian Li

**Affiliations:** 1School of Civil and Hydraulic Engineering, Ningxia University, Yinchuan 750021, China; lxqiangyx1301@163.com (X.L.); siqi7li@163.com (S.L.); qiangxiaolin123@163.com (X.Q.); nanxia_yu@163.com (Z.Y.); 2China-Arab Joint International Research Laboratory for Featured Resources and Environmental Governance in Arid Region, Yinchuan 750021, China; hejun3025@163.com (J.H.); layhan@163.com (L.H.); li_q@nxu.edu.cn (Q.L.); 3Key Laboratory of Resource Assessment and Environmental Control in Arid Region of Ningxia, Yinchuan 750021, China; 4School of Mechanical Engineering, Ningxia Vocational Technical College of Industry & Commerce, Yinchuan 750021, China; agridrizzle@163.com

**Keywords:** apple trees, fruit quality, logistic growth model, subsurface infiltration irrigation, water and nitrogen regulation, water and nitrogen supply decision-making

## Abstract

Apple tree productivity is influenced by the quantity of water and nutrients that are supplied during planting. To enhance resource utilization efficiency and optimize yields, a suitable strategy for supplying water and nitrogen must be established. A field experiment was conducted using a randomized block group design on five-year-old apple trees in Ningxia, with two irrigation lower limit levels (55%FC (W1) and 75%FC (W2)) and four N application levels (0 (N1), 120 (N2), 240 (N3), and 360 (N4) kg·ha^−1^). Our findings showed that leaf N content increased with a higher irrigation lower limit, but the difference was not statistically significant. However, the leaf N content significantly increased with increasing N application. The growth pattern of new shoots followed logistic curve characteristics, with the maximum new shoot growth rate and time of new shoot growth being delayed under high water and high nitrogen treatments. Apple yield and yield components (weight per fruit and number of fruits per plant) were enhanced under N application compared to no N application. The maximum apple yields were 19,405.3 kg·ha^−1^ (2022) and 29,607 kg·ha^−1^ (2023) at the N3 level. A parabolic relationship was observed between apple yield and N application level, with the optimal range of N application being 230–260 kg⸱ha^−1^. Apple quality indicators were not significantly affected by the irrigation lower limit but were significantly influenced by N application levels. The lower limit of irrigation did not have a significant impact on the quality indicators of the apples. Water and N utilization efficiencies improved with the W2 treatment at the same N application level. A negative relationship was observed between the amount of nitrogen applied and the biased productivity of nitrogen fertilizer. The utilization of nitrogen fertilizer was 127.6 kg·kg^−1^ (2022) and 200.3 kg·kg^−1^ (2023) in the W2N2 treatment. The apple yield was sustained, the quality of the fruit improved, and a substantial increase in water productivity was achieved with the W2N3 treatment. The findings of this study can be used as a reference for accurate field irrigation.

## 1. Introduction

Water resources are becoming increasingly scarce in many arid and semiarid regions [1]. Irrigation for food production consumes most of the world’s freshwater, accounting for approximately 70% of the total water resources [2]. However, crops effectively use less than 60% of the irrigation water [3]. The proportion of economic forestry and fruit cultivation in agricultural planting systems has been increasing annually, in tandem with social development. By 2023, the apple planting area will have reached 2.23 × 10^6^ ha, with an annual fresh fruit production of 41.39 × 10^6^ t, accounting for 45.2% of the world’s planting area and 49.87% of global fresh fruit production [4]. Apple yield and biomass are higher than those of conventional annual crops, making water and nutrients critical for efficient and high-quality apple production. A lack of scientific guidance, traditional farmer beliefs, and the combined impact of soil and production environments result in inefficient water and fertilizer usage in orchards. This leads to economic losses for fruit farmers and several environmental challenges, including nitrogen leaching and volatilization [5]. Considering the importance of water-nitrogen interactions, water and nitrogen management in fruit tree production processes require further in-depth research.

Crop water physiology is a complex and systematic phenomenon. The crop root system absorbs water and transports it to various above-ground organs, allowing them to maintain their normal physiological functions. Water is not just a solvent for nutrient absorption; it is also crucial for crop growth. Nitrogen is essential for crop growth and development and regulates the growth of roots, stems, leaves, and fruits. Nitrogen can contribute to up to 40–50% of the final crop yield, making it a critical factor limiting crop production [6]. Therefore, water and nitrogen are complementary and indispensable for crop production. Previous research has shown that high-water and high-fertilizer treatments can increase yield by up to 139% compared to low-water and low-fertilizer treatments, improving partial fertilizer productivity while decreasing water productivity and fruit quality [7,8]. Furthermore, several researchers have discovered that individual fruit weight is positively correlated with the amount of nitrogen fertilizer applied [9]. Other studies have indicated that increasing fertilizer application under reduced water conditions significantly affects fruit quality, explicitly reducing aromatic compound content while increasing soluble solids, crispness, and hardness [10]. Nitrogen has the most significant effect on fruit hardness compared to water, and its effect on fruit color ranks third for potassium and phosphorus [11]. As the amount of applied nitrogen increases, the accumulation of soluble solids in the fruit also increases, whereas the hardness of the flesh decreases [12]. Therefore, there is an urgent need to explore a reasonable water-nitrogen supply strategy to maintain or improve apple yield and quality and satisfy the needs of fruit growers and consumers.

The timing of water and nitrogen supply during the phenological stages of apple trees is also a crucial factor that affects yield and quality. High nitrogen topdressing in the summer did not improve overall fruit quality. Instead, it reduces fruit firmness at harvest while increasing acidity, resulting in a lower yield. Applying nitrogen 8–12 weeks after flowering reduces starch content and increases soluble solid content, which improves fruit maturity but has little effect on coloration [12]. The nutritional status of fruit trees varies significantly depending on water and nitrogen supply strategies, resulting in distinct growth and development characteristics. Studies using isotope N15 labeling experiments have determined that the critical period for dry matter and nitrogen accumulation in leaves and new shoots is 30–60 days after bud break. However, the crucial period for dry matter and nitrogen accumulation in fruits is 120–180 days. Tree nitrogen fertilizer use efficiency was also the highest during these two periods [13]. Increasing the fertilizer application (N, P, and K) while maintaining constant water conditions increased the soluble solid content, sugar content, and fruit hardness of apples. Conversely, increasing irrigation while keeping the fertilizer application constant reduced the titratable acid content of apples and increased the sugar-acid ratio. Fertilization has no significant effect on this [10,14].

In agricultural research, growth models are frequently used to characterize the relationship between crop growth and environmental factors. The logistic model is the most commonly employed growth model and has demonstrated excellent performance in simulating dynamic crop growth, as evidenced in several studies [15,16]. It has been used to describe the growth of tomatoes [17], peppers [18], and fruit trees [19], as well as that of winter wheat, summer maize, and sugar beets [15]. However, there is still a need for more conclusive studies linking water and fertilizer-use efficiency indicators.

In summary, research findings on the responses of apple growth, development, yield, and quality to water and fertilizer availability have been inconsistent. These discrepancies are related to the frequency and total amount of irrigation and nitrogen application, as well as the phenological stages of the fruit trees. Therefore, it is essential to systematically regulate water and fertilizer management techniques for high-quality and efficient apple production. Our study evaluated crop sensitivity to water and nitrogen during critical periods and determined yield and quality indicator reaction thresholds for soil moisture and nitrogen application. Furthermore, we used subsurface drip irrigation technology to supply water and nutrients to crops in a synchronous, even, timely, quantitative, and precise manner, resulting in the synchronous coupling of water and nutrients in time and space. This approach could improve the coordination and coupling effects of water and fertilizer supply in crop production, significantly increasing irrigation water and fertilizer utilization efficiency. Our study provides a scientific basis and guidelines for achieving high-quality and efficient apple production by controlling water and nitrogen levels.

## 2. Results

### 2.1. Response of Leaf Nitrogen Content to Water and Nitrogen Regulation in Apple Trees

The trends in the dynamics of leaf N content were essentially the same during the two-year growing season of apple trees from 2022–2023 (Figure 1). The leaf N content reached its highest level during the flowering stage. As the reproductive period progressed, the leaf N content gradually declined, falling to its lowest point at the late stage of fruit expansion. Leaf N content rebounded during fruit coloring but declined again after fruit harvest. Leaf N content was higher in the W2 treatment than in the W1 treatment, suggesting that increasing the lower limit of irrigation (increasing soil water content) had a positive but non-significant (*p* > 0.05) effect on leaf N content. Nitrogen application treatments (N2, N3, and N4) had higher leaf N contents than the non-applied treatments (N1). The higher nitrogen application treatment N4 (360 kg⸱ha^−1^) had a significant effect (*p* < 0.05) on the increase in leaf N content, whereas there was no significant difference in leaf N content between the N2 (120 kg⸱ha^−1^) and N3 (240 kg⸱ha^−1^) treatments (*p* > 0.05). The highest relative rate of change in leaf N content was observed in the non-applied nitrogen treatment (N1).

The ANOVA results (Table 1) showed that in 2022, soil moisture treatments significantly (*p* < 0.05) affected leaf N content at the young fruiting stage, whereas they did not significantly affect leaf N content at the rest of the reproductive stages. Nitrogen application treatments had no significant effect (*p* < 0.01) on leaf N content at the flowering stage but reached a highly significant level (*p* < 0.01) for the rest of the reproductive stages. The interaction effect of soil moisture and nitrogen application had a significant effect on leaf N content at anthesis, but not at the rest of the fertility stages. In 2023, soil moisture treatments had a significant effect on leaf N content during the rapid fruit expansion stage and a highly significant effect (*p* < 0.01) on leaf N content during the rest of the reproductive stages. Nitrogen application had highly significant (*p* < 0.01) effects on leaf N content at all reproductive stages in apple trees. The interaction effect of soil moisture and nitrogen application also had a significant effect on the leaf N content of apple trees at the full reproductive stage.

### 2.2. Growth Status of Apple Tree New Shoots (Spring Shoots) in Response to Water and Nitrogen Regulation

Based on observations of new shoot growth (spring shoots) and growth rate, it was demonstrated that the growth pattern of new shoots conformed to the characteristics of the logistic curve (S-shaped curve equation). The logistic curve fit well under the different treatments (Figure 2), with determination coefficients (R^2^) ranging from 0.928 to 0.994 (W1) and 0.994 to 0.999 (W2). A logistic model was used to calculate the characteristic parameters of new shoot growth (Table 2). It was observed that in the W2 treatment, the simulated value of the maximum new shoot growth rate was slightly delayed compared to the W1 treatment (except under the N2 condition in 2022), indicating that the W2 treatment extended the duration of new shoot growth compared to the W1 treatment. During the rapid growth period, the average duration (T_m_) and the time the growth rate reached its maximum value (t_m_) increased with an increase in the lower limit of irrigation, with the W2 treatment being higher than the W1 treatment. However, nitrogen application treatments (N2, N3, N4) delayed the occurrence of the maximum new shoot growth rate compared to the no nitrogen application (N1) treatment while prolonging the duration of new shoot growth, as was especially evident in the results of the 2023 experiment. The growth rate (v_m_) of the W1 treatment was higher than that of the W2 treatment, reaching a maximum of 1.24 cm·d^−1^ under the N3 condition in 2022 and 1.01 cm·d^−1^ under the N4 condition in 2023.

### 2.3. Response of Intrinsic Quality of Apple to Water and Nitrogen Regulation

According to the results of experiments conducted between 2022 and 2023 (Figure 3 and Figure 4), increasing soil moisture and nitrogen application will decrease flesh firmness (FF) (*p* < 0.05), but the difference was not significant under the same nitrogen treatment. A lower irrigation threshold and nitrogen application will increase the total soluble solid (TSS) content of apples. The TSS content of apples under treatment W1 was higher than that under treatment W2, but the difference was not statistically significant. Under the nitrogen application treatments (N2, N3, and N4), the TSS content of apples was significantly higher than that of apples without nitrogen application (N1) (*p* < 0.05). The TSS content gradually stabilized after nitrogen fertilizer increased from N3 to N4. Reducing soil moisture increased the soluble sugar content (SSC) of apples, and nitrogen application significantly affected SSC (*p* < 0.05). Reducing soil moisture can decrease the TA content of apples, and nitrogen application substantially affects the titratable acidity (TA) content of apples. The TA content decreased slightly and then increased slightly with increasing nitrogen application, reaching a minimum at the N3 level. The irrigation gradient and nitrogen application significantly affected the vitamin C (VC) content of the apples (*p* < 0.05). Increasing nitrogen fertilizer significantly increased VC content (from N1 to N2) (*p* < 0.05), whereas VC content increased slightly from N2 to N4 levels but not significantly (*p* > 0.05). Trends related to changes in soil water content and nitrogen usage for the sugar-acid ratio (SSC/TA) were similar. Specifically, lower soil moisture levels were found to increase SSC/TA, whereas higher levels of nitrogen application also resulted in increased SSC/TA. In addition, the SSC/TA ratio peaked at the N3 level, which was significantly higher than that observed without nitrogen application (*p* < 0.05).

The results of the correlation analysis conducted on the internal quality indicators of apples (Figure 5) indicated a strong positive correlation between TSS and SSC, as well as VC, while exhibiting a strikingly negative correlation with FF and TA. Specifically, VC demonstrated a remarkably positive correlation with TSS, SSC, and SSC/TA and a significant negative correlation with FF and TA. Additionally, FF showed a pronounced negative correlation with TSS, SSC, and VC, a moderately positive correlation with SSC/TA, and an insignificant correlation with TA.

### 2.4. Response of Yield and Its Components to Water and Nitrogen Regulation

The results of the experiments conducted from 2022 to 2023 (Figure 6) indicate that the single fruit weight (SFW) is significantly higher under nitrogen application treatments (N2, N3, N4) than in the absence of nitrogen application treatment (N1). Additionally, the number of fruits per plant (FNP) peaked in the N3 treatment under identical irrigation gradients. Moreover, under consistent nitrogen application conditions, the apple yield (Y) was more significant in the W2 treatment than in the W1 treatment. Across the two irrigation gradient conditions, apple yield increased with nitrogen application, notably peaking in the N3 treatment, with yields of 18,150.7 kg·ha^−1^ and 19,405.3 kg·ha^−1^ in 2022 and 28,759 kg·ha^−1^ and 29,607 kg·ha^−1^ in 2023, significantly surpassing the no nitrogen application treatment (*p* < 0.05). Under combined water and nitrogen regulation, the W2N3 treatment achieved the maximum SFW, number of FNP, and yield (Y), highlighting the beneficial effects of optimal water and nitrogen inputs on enhancing apple yield and its constituent elements. A comprehensive analysis of the two-year experimental data revealed that apple tree productivity in 2023 exceeded that in 2022.

### 2.5. Effect of Water Consumption on Water Productivity and Yield of Apple Trees

By analyzing the effects of water consumption (ET) during the growing season of apple trees from 2022 to 2023 on yield (Y) and water productivity (WP), polynomial regression equations were derived, with yield and water productivity serving as dependent variables and water consumption as the independent variable (Figure 7). The graph indicates that as water consumption increases, both yield and water use efficiency initially experience an increase, followed by a decrease, with the rate of growth exceeding that of the decline. Beyond a certain threshold of water consumption, the yield and water productivity decrease. Compared with the W1 irrigation treatment, W2 exhibited improved water productivity. In 2023, water productivity significantly surpassed that in 2022 (*p* < 0.05).

### 2.6. Effect of Nitrogen Application on Nitrogen Partial Factor Productivity and Yield of Apple Trees

By evaluating the impact of nitrogen application (N) during the phenological period of apple trees from 2022 to 2023 on nitrogen partial factor productivity (NPFP) and yield (Y), regression equations for N-NPFP and N-Y were derived, with N serving as the independent variable and NPFP and Y as dependent variables (Figure 8). NPFP demonstrated a linear regression relationship with nitrogen application, suggesting that moderate nitrogen application can enhance NPFP. However, NPFP significantly decreased with increasing nitrogen application (*p* < 0.05). The NPFP in the W2 treatment was greater than that in the W1 treatment, indicating that increasing the irrigation threshold under appropriate nitrogen application levels can improve NPFP, whereas excessive nitrogen application significantly reduces NPFP. Apple yield followed a polynomial regression curve with varying nitrogen applications, displaying an initial increase followed by a decrease, with the highest yield occurring in the N3 treatment. Upon comprehensive analysis of the two-year experimental data, the W2N3 treatment was found to have the highest NPFP, reaching 80.9 kg·kg^−1^ in 2022 and 123.4 kg·kg^−1^ in 2023, and both NPFP and yield in 2023 surpassed those in 2022.

## 3. Discussion

### 3.1. Effects of Water and Nitrogen Regulation on Leaf Nitrogen Content and Growth of Apple Trees

The leaf nitrogen content of apple trees changes with nitrogen application levels, potentially resulting in distinct leaf color variations owing to excessive or inadequate nitrogen application [20,21]. Furthermore, leaf nitrogen content fluctuated throughout the phenological stages of the fruit trees. Leaf nitrogen concentration often gradually decreases from flowering to late fruit expansion, probably because of the greater need for nutrients to support vigorous growth and development during this period. However, once the trees reach the coloring stage, their physiological demand for nutrients decreases, increasing leaf nitrogen content. Following fruit harvest, leaf nitrogen levels drop again as trees reallocate nitrogen to other organs for storage, preparing for the following year [22]. In our study, the nitrogen content in the leaves was the highest under the high-nitrogen treatment (N4, 360 kg·ha^−1^), which was significantly higher than that under the no-nitrogen treatment (N1). However, there was no significant difference compared with the other nitrogen treatments (N2 and N3). This could be because the high-nitrogen treatment increased the available nitrogen in the root zone beyond the tree’s requirements; however, the tree could not fully assimilate excess nitrogen [12]. In addition, this study found that the 2022 trial had a lower leaf nitrogen content than the 2023 trial. Pruning procedures adopted in 2022 to manage the main branches and promote sufficient flower bud formation [23,24] may have contributed to the enhanced yield in 2023.

The growth of apple tree canopies is influenced by water and nitrogen supply, with an adequate supply promoting shoot growth. This study discovered a significant reduction in shoot length due to nitrogen deficiency, highlighting the importance of nitrogen availability for shoot growth. Interestingly, the high-nitrogen treatment in this study did not result in longer shoots, which could be because the high-nitrogen treatment did not foster the growth of longer spring shoots. This could be partially attributed to this study’s emphasis on measuring long, medium, and short shoots rather than extension shoots, which are crucial for flower bud formation and serve as carbon sources for apple trees during the swelling period [8]. Plant hormones may also affect shoot growth. Spring and autumn shoot growth were significant in 2022, but autumn shoot growth was limited in 2023, possibly because of an altered fruit load between the two years. Because fruit growth is considered the primary source of carbon sequestration in apple trees, more photosynthetic products are directed towards shoot growth in years when fruit load is low. However, in years with large fruit loads, most photosynthetic products are allocated to fruit growth, which inhibits shoot growth, particularly in autumn [8].

### 3.2. Impact of Water and Nitrogen Regulation on Apple Yield and Quality

Irrigation and nitrogen application are the primary factors that influence apple yield [25]. Our study showed that fluctuations in nitrogen fertilizer supply significantly affected apple productivity. Apple yield exhibited a pattern of increase followed by a drop as nitrogen application increased under the two distinct irrigation gradients. Within the 0–240 kg·ha^−1^ range, apple yield and fruit number increased significantly as nitrogen application increased. However, when nitrogen application reached 360 kg·ha^−1^, fruit number and yield declined. This implies that increasing water may boost yield under limited water and fertilizer supply or in poor soil, but that increased fertilizer application has a more significant impact. Moderate nitrogen application can accelerate tree growth, enhance water and nutrient uptake, facilitate photosynthesis assimilation, improve flower bud differentiation quality and fruit set rate, and increase fruit number and yield [26]. Nevertheless, excessive nitrogen fertilizer can disrupt the carbon-nitrogen equilibrium, inhibit the absorption of other micronutrients, reduce carbohydrate proportions, and cause nutrient-element ratio imbalances [27]. Moreover, excess nitrogen can trigger excessive gibberellin production in plants, suppressing ethylene production, flower bud differentiation, and fruit set rate, resulting in a lower yield [28]. We observed that the apple yield at the N4 level was lower than that at the N3 level, supporting the idea that excessive nitrogen application can reduce yield. Hence, nitrogen fertilizer should be administered sparingly, depending on soil fertility and actual plant requirements, to mitigate the adverse effects of excessive nitrogen application.

The application of nitrogen and irrigation affects apple quality. FF initially decreases with increased nitrogen application and irrigation before stabilizing, possibly because of the soil and plant characteristics and the potassium and calcium content in the fruit [23,29]. While SSC increased with increasing nitrogen application, TSS and VC exhibited an initial increase, followed by a drop. Furthermore, under identical treatment conditions, the TSS of apples in 2023 was lower than that in 2022, probably because of dilution due to the increased yield. When nitrogen application increased, the TA content initially declined and then increased slightly, most likely because of the nitrogen-to-potassium nutrient ratio in the soil [30,31]. An analysis of Apple’s intrinsic quality indicators revealed a strong correlation between them. TSS and SSC were negatively correlated with TA content, probably because of the conversion of acids to sugars during apple ripening [32]. We found that VC was significantly and positively correlated with TSS and SSC, as SSC promotes VC synthesis and serves as its primary substrate [33]. These findings can help us understand the complex mechanisms that govern apple quality and provide crucial insights for optimizing apple horticulture management.

### 3.3. Effects of Water and Nitrogen Regulation on Water Consumption and Nitrogen Utilization in Apple Trees

Crop water consumption patterns are critical to agricultural productivity and are influenced by various parameters, including soil conditions, irrigation systems, and meteorological factors [34]. Water consumption in apple trees typically increases initially before declining, with the fruit swelling period marking the highest water consumption peak and the budding period the lowest. Possible causes for this phenomenon include the high physiological activity of apple trees during the fruit swelling period, which is characterized by peak growth and development, elevated temperatures, and a high leaf area index, consequently leading to a surge in water consumption [35]. Increased irrigation and nitrogen application promote apple tree growth and development, enhance the leaf area index (LAI), and increase water consumption [36]. Thus, improving the lower threshold of irrigation and nitrogen application can bolster the growth and development of apple trees, thereby augmenting their water consumption. This offers significant insights for water resource planning and management in agriculture and the operation of farmland irrigation and drainage systems.

Increased water consumption reduces apple productivity. Under consistent nitrogen application levels, increasing the upper limit of irrigation improved water utilization efficiency, albeit insignificantly (*p* > 0.05). Consistent with an earlier study by Liao et al. [26] on various degrees of deficit irrigation in apple orchards, the results from 2022 and 2023 showed that maximal water productivity occurred under high irrigation lower limits (W2 75%FC) and N3 (240 kg·ha^−1^) treatments. Conversely, nitrogen fertilizer use efficiency declined with increasing nitrogen application. According to the law of diminishing marginal returns, this might be due to heightened sensitivity to nitrogen under low-nitrogen conditions as opposed to high-nitrogen conditions [37]. In addition, excessive nitrogen application can result in elevated tree load and aggressive vegetative growth, inhibiting adequate nutrient supply for reproductive growth, affecting root activity, reducing nitrogen absorption, and ultimately diminishing N-use efficiency [38]. These findings highlight the significance of adequate nitrogen application control for maintaining water productivity in apple trees and maximizing agricultural output.

## 4. Material and Methods

### 4.1. Experiment Location

The field experiment was conducted from October 2021 to October 2023 at the Ningxia Dryland Water-saving High-efficiency Agricultural Science and Technology Park Experimental Station (36°50′ N, 105°60′ E; Wuzhong, Ningxia, China) (Figure 9). The mean annual temperature in the experimental region is 8.6 °C. The mean annual precipitation is 270 mm, with rainfall primarily concentrated from July to September, leading to an uneven distribution throughout the year. The mean annual evaporation in the experimental region was 2325 mm. The duration of the frost-free period varied between 120 and 218 days, while the average annual duration of sunshine was 3024 h. In the autumn of 2021, the physical and chemical properties of soils in the 0–60 cm layer were measured. The measurements showed an alkaline hydrolyzed nitrogen content of 43.46 mg·kg^−1^, an available phosphorus content of 21.53 mg·kg^−1^, an available potassium content of 114.17 mg·kg^−1^, an organic matter content of 12.84 g·kg^−1^, a bulk density of 1.51 g·cm^−3^ a pH level of 7.68, a field capacity of 24.8% (volumetric), and a permanent wilting point of 8.5%. The soil in the experimental area was classified as sandy loam based on its texture.

### 4.2. Experimental Design

A field experiment was conducted using a randomized block group design on five-year-old apple trees in Ningxia, with two irrigation lower limit levels (55%FC (W1) and 75%FC (W2)) [26] and four N application levels (0 (N1), 120 (N2), 240 (N3), and 360 (N4) kg·ha^−1^) [37]. The irrigation threshold utilized in this study was the field capacity (FC) of the 0–60 cm soil depth in the experimental area. In total, there were eight treatments (Table 3). Three fruit trees exhibiting comparable growth were chosen for each treatment, and each treatment was replicated three times to minimize plot effects. The maximum irrigation threshold was 90%FC for all the treatments. The amounts of phosphorus fertilizer (P_2_O_5_) and potassium fertilizer (K_2_O) were consistent across all treatments. P_2_O_5_ was applied at a rate of 225 kg⸱ha^−1^, and K_2_O was applied at a rate of 300 kg⸱ha^−1^ from 2021–2022. P_2_O_5_ was applied at a rate of 195 kg⸱ha^−1^, and K_2_O was applied at a rate of 240 kg⸱ha^−1^ from 2022–2023. The precise fertilizer regimes are listed in Table 4. Urea (N 46%) was used as a nitrogen fertilizer, and potassium dihydrogen phosphate (P_2_O_5_ 52%, K_2_O 34%) was used as a potash fertilizer in the experiment. Potassium chloride was used as a supplement to compensate for the insufficient use of potash fertilizers.

The apple trees in the experimental area were of the “Huashuo” variety, which belongs to early maturation. These trees were grafted onto the M26 dwarf rootstock as an intermediate rootstock, with the base rootstock being the malus seedling rootstock. The trees were aligned in a north-south direction. The spacing between planting rows was 200 × 300 cm, resulting in a planting density of 1600 trees per hectare. The tree trunk had a slender, spindle-shaped form and measured approximately 290 cm in height. The width of the tree crown was 164 cm from north to south and 157 cm from east to west. The fruit matured from mid-August to late-August. The fruit can be stored for approximately 30 days at room temperature without softening. Furthermore, it may be preserved for three months under refrigerated conditions. The fruit can be harvested either prematurely or in multiple stages once it reaches full maturity [39].

**Table 4 plants-13-02404-t004:** Nutrient input ratios during each growth period. The data in the table are the proportion of fertilizer applied during the corresponding fertility period to the total amount used in the year.

Growth Period	Time	Experiment Year					
		2021–2022			2022–2023		
		N	P_2_O_5_	K_2_O	N	P_2_O_5_	K_2_O
		%	%	%	%	%	%
After harvest (base fertilizer)	early-October	20	16	8	20	16	8
Budding stage	late-March	30	31	15	30	31	15
New shoot growth stage	mid-April	10	16	8	10	16	8
Flowering stage	early-May	20	12	9	20	12	9
Fruit setting	early-June	10	10	10	10	10	10
Early fruit enlargement stage	late-June	10	5	10	10	5	10
Rapid fruit enlargement stage	mid-July	0	5	20	0	5	20
Later fruit enlargement stage	early-August	0	5	20	0	5	20

Note: According to the BBCH scale coding record, the main growth period of apple trees is divided into the budding stage (09); the new shoot growth stage (3) includes the beginning of new shoot growth (31); about 30% of the final length (33); about 50% of the final length (35); about 70% of the final length (37); and about 90% of the final length (39); flowering stage (65); fruit development stage (7) includes fruit setting (71); early fruit enlargement stage (72); rapid fruit enlargement stage (74); late fruit enlargement stage (79); fruit maturity stage (8), including coloring stage (81); and maturity stage (89) [26,40].

A subsurface infiltration irrigation system was used for irrigation. The infiltration irrigation pipes were positioned in a circular formation surrounding the tree trunk (radius R = 30 cm; burial depth = 20 cm). Root-drilling sampling of the experimental apple trees revealed that more than 80% of the roots were located within the soil layer, ranging from 0 to 60 cm. The microporous rubber infiltration irrigation pipes had an outer diameter of φ22, an inner diameter of φ16, and a flow rate of 12 L∙m^−1^h^−1^. The upper and lower limits of irrigation were controlled using a mobile intelligent control system (GC-003, ANC Technology Co., Shanghai, China). Irrigation time, irrigation amount, and soil moisture changes were updated and logged every 5 min during the irrigation process. When irrigation ceased, the data were updated and logged every 30 min. Based on the distribution of apple tree roots, three sensors were deployed for each treatment (calibrated to suit the field conditions) [41]. The sensors were buried at depths of 10, 30, and 50 cm (Figure 10). The cumulative irrigation volumes of the two lower-limit irrigation treatments for apple tree phenology were counted at the end of the experiment (Figure 11).

### 4.3. Plant Measurements

#### 4.3.1. Measurement of New Shoot (Spring) Growth

Based on the annual growth of apple trees, the growth of spring shoots in apple trees was observed at the early stage of spring shoot growth. Three apple trees exhibiting comparable development were selected and marked for each treatment. Three representative main branches were chosen from the east, south, west, and north directions of each marked apple tree, for a total of 12 branches. The plants were sequentially labeled with plates indicating their order, and the length of the spring shoots on the main branches (excluding long branches) was measured using a steel ruler with a precision of 1 mm. The duration of the measuring cycle was approximately 10 days, and monitoring was sustained until the growth of the spring shoots ceased.

#### 4.3.2. Measurement of Leaf Nitrogen Content

Leaf samples from the apple trees were collected at different stages of fertility. For each treatment, six current-year shoots were selected from the upper and lower parts and around the canopy of the marked apple trees. Four healthy, undamaged, and mature leaves (without petioles) were collected from each shoot and placed in a single preservation bag. The labeled bags were placed in a cooler with ice packs and brought back to the laboratory [42]. Leaves were washed with distilled water to remove surface dust and fuzziness from the underside. The leaves were dried in a forced-air oven at 85 °C for 30 min. Then, at 70 °C, they reached a constant weight. Once cooled, the samples were ground into powder using a mortar and pestle. The ground leaves were passed through a 0.45 mm sieve and sealed for later use. A 0.2 g sample was placed into a digestion tube, and 5 mL of concentrated sulfuric acid (H_2_SO_4_) was added. The digestion tube was heated to 380 °C with the repeated addition of hydrogen peroxide (H_2_O_2_) until the solution became clear. After cooling, the digested solution was transferred to a 100 mL volumetric flask and diluted to volume. After mixing, 15 mL was removed using a pipette, and the nitrogen content of the leaves was determined using the Kjeldahl method (K1100F) [43]. The calculation formula is as follows:(1)$$N=\rho \times V\times ts\times 10^{-3}/m$$
where *N* is the leaf nitrogen content (g·kg^−1^), *ρ* is the mass concentration of the color-developing liquid *N* (4${NH}_{4}^{+}$-N) from the standard curve (μg⸱mL^−1^), *V* is the volume of the color-developing liquid (mL), *ts* is the splitting multiplier, is the volume of the decoction liquid (mL)/volume of the absorbed decoction liquid (mL), *m* is the weighed mass of the dry matter sample (g), and 10^−3^ is the conversion coefficient of the unit.

#### 4.3.3. Measurement of Yield and Its Components

During the fruit maturity and harvest periods, the field weighing method was used to harvest the marked apple trees according to the different treatments. The yield per tree was measured using an electronic balance (0.01 g) (Precisa XJ3200C, Shanghai, China), and the number of fruits per tree and the weight of individual fruits were recorded. The total yield per hectare for each treatment was calculated based on planting density.

#### 4.3.4. Measurement of Fruit Quality Indicators

During the fruit maturity and harvest periods, three representative fruits were selected from the east, west, south, and north of each tree, for a total of 12 fruits per treatment. The harvested fruit samples were placed in preservation boxes and taken back to the laboratory, where they were stored in a refrigerator at 5 °C. Quality analysis was completed within one week, employing a GY-4 digital hardness tester to gauge fruit flesh hardness; the anthrone-sulfuric acid colorimetric method was used to ascertain the soluble sugar content in the fruits; a handheld refractometer was utilized to gauge the soluble solid content in the fruits; the vitamin C content in the fruits was determined using the molybdenum blue colorimetric method; and the titration method with sodium hydroxide was employed to measure the titratable acidity content in the fruits [44].

### 4.4. Calculation of Water Consumption

The total water consumption (ET) and water consumption at a specific growth stage were calculated using the following equations [45]:(2)ET=P+U+I−D−R−∆W
where ET is crop evapotranspiration, *P* is precipitation, *U* is groundwater recharge, *I* is the amount of irrigation, *D* is deep drainage, *R* is surface runoff, and Δ*W* is the soil moisture change from the beginning to the end of the experiment [46]. In the experimental field, the groundwater depth was below 16 m, and the observable groundwater recharge was considered zero (*U* = 0). Additionally, the depth of precipitation infiltration did not exceed 2 m, rendering observable deep leaching of zero (*R* = 0). Slight fluctuations or a constant decrease in soil moisture in the deep layers indicate that no deep drainage occurred (*D* = 0).

### 4.5. Calculation of Water and Nitrogen Utilization Efficiency

(1)Water productivity (WP) [47]

(3)WP=Y/10ET
where WP is the water productivity (kg·m^−3^), *Y* is the apple yield (kg⸱ha^−1^), and ET is the water consumption during the reproductive period (mm).

(2)Nitrogen partial factor productivity (NPFP) [48]

(4)NPFP=Y/F
where NPFP is the nitrogen partial factor productivity (kg·kg^−1^), *Y* is the apple yield (kg⸱ha^−1^), and *F* is the nitrogen application (kg·ha^−1^).

### 4.6. Model Description and Application

Logistic equations were used to quantify the effects of water and nitrogen supply on the growth of apple trees, which are described as follows [15]:(5)y=k/(1+a⁡×EXP⁡(b⁡×t⁡))
where *y* is the growth parameter (new growth, cm) of the apple trees in this study, *k* is the uppermost asymptote indicating the upper limit of new growth (cm), *a* and *b* are the initial stage and accretion rate coefficients, respectively, and *t* is the independent elapsed time (d).

The characteristic parameters of the Logistic Equation are calculated as follows:(6)$$v_{m}=bk/4$$
(7)$$t_{m}=(\ln a)/b$$
(8)$$t_{1}=\left(\ln(2+\sqrt{3)}/a\right)/b$$
(9)$$t_{2}=\left(\ln(2-\sqrt{3)}/a\right)/b$$
(10)$$T_{m}=t_{2}-t_{1}$$
where *v_m_* is the maximum growth rate during the rapid growth period, *t_m_* is the time node corresponding to the time when the growth rate reaches the maximum value (*v_m_*), *t*_1_ and *t*_2_ are the inflection points of the logistic equation, which are the beginning and ending time nodes of the rapid growth period, respectively, and *T_m_* denotes the duration of the rapid growth period (d).

### 4.7. Meteorological Factors

Rainfall, air temperature, atmospheric pressure, relative air humidity, solar radiation, wind velocity, and wind direction were obtained at a height of 2.0 m using an automatic weather station installed in the orchard (Figure 12). Rainfall greater than 5 mm during a single event was considered adequate. The FAO Penman-Monteith equation calculates the reference evapotranspiration (ET0) [49].

### 4.8. Statistical Analysis and Computation

An analysis of variance (ANOVA) was performed using SPSS 22.0 software of IBM (New York, NY, USA). Duncan’s multiple range test was used to determine whether there were significant differences among all treatment methods (*p* = 0.05). SPSS software (version 22.0) was used for the correlation analysis. Origin 2021 was used for logistic, linear, and polynomial fittings.

## 5. Conclusions

This study aimed to investigate the response of apple trees to various water and nitrogen regulation conditions in terms of nitrogen nutrition, canopy growth characteristics, yield quality, and water-nitrogen use efficiency. The results showed that at the same nitrogen application level, the W2 (75%FC) treatment had a higher leaf nitrogen content than the W1 (55%FC) treatment, although the difference was not significant (*p* > 0.05). Nitrogen administration significantly improved leaf nitrogen content compared to the no-nitrogen (N1) treatment (*p* < 0.05). The growth patterns of the new shoots followed a logistic growth curve. High-water and high-nitrogen treatments delayed the maximum new shoot growth rate and extended the growth period of the new shoots. Apple tree yields at the N3 level were highest at 19,405.3 kg·ha^−1^ (2022) and 29,607 kg·ha^−1^ (2023). Apple yield was found to have a parabolic relationship with nitrogen application rate, with 230–260 kg·ha^−1^ being the theoretically optimal range for nitrogen application. The nitrogen application rate significantly affected the quality indices of the apples, but the lower irrigation limit had no significant effect (*p* > 0.05). At the same nitrogen application level, the W2 treatment improved the utilization efficiency of water and nitrogen fertilizers. The amount of nitrogen fertilizer negatively affected the partial productivity of nitrogen fertilizer. The W2N2 treatment resulted in agronomic nitrogen fertilizer efficiency of 127.6 kg·kg^−1^ (2022) and 200.3 kg·kg^−1^ (2023), while the W2N3 treatment improved fruit quality and water productivity significantly while maintaining apple yield.

## Figures and Tables

**Figure 1 plants-13-02404-f001:**
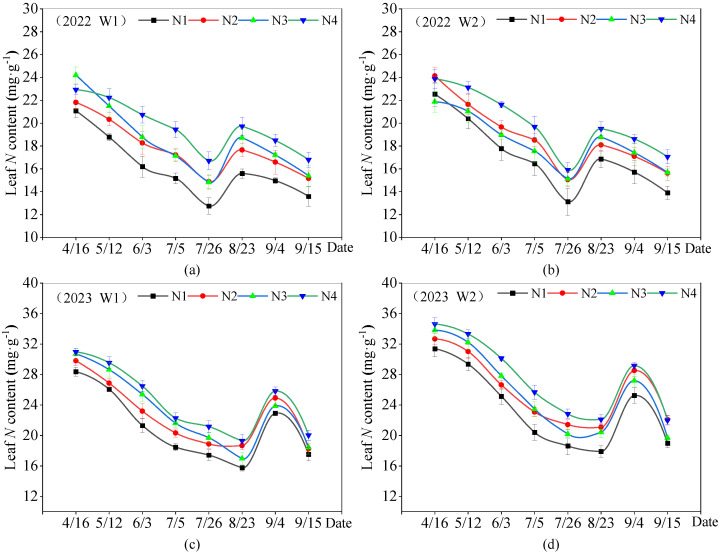
Dynamics of changes in leaf nitrogen content of apple trees in 2022–2023. Dotted line plots represent mean ± standard deviation (n = 3). Note: (**a**) is leaf *N* content on apples trees under W1 treatment conditions in 2022, (**b**) is leaf *N* content on apples trees under W2 treatment conditions in 2022, (**c**) is leaf *N* content on apples trees under W1 treatment conditions in 2023, (**d**) is leaf *N* content on apples trees under W2 treatment conditions in 2023.

**Figure 2 plants-13-02404-f002:**
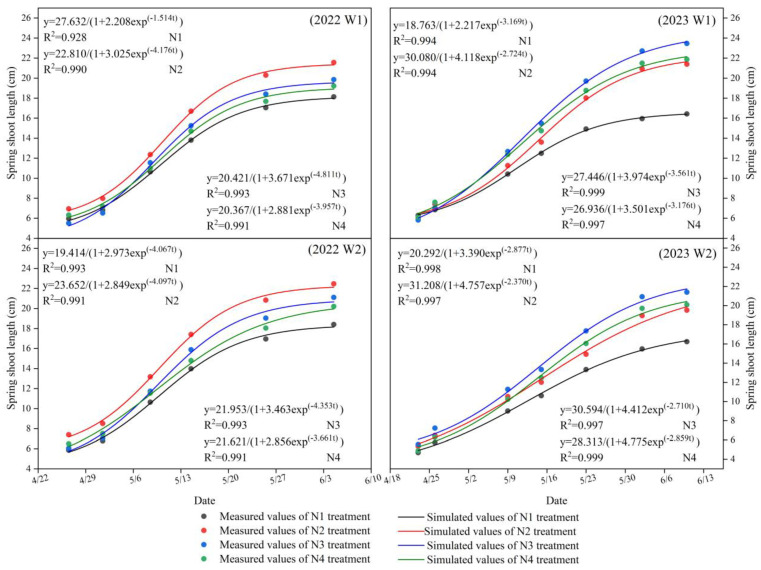
Spring shoot growth conditions of apple trees in response to water and nitrogen regulation in 2022–2023. All samples were fertility observations.

**Figure 3 plants-13-02404-f003:**
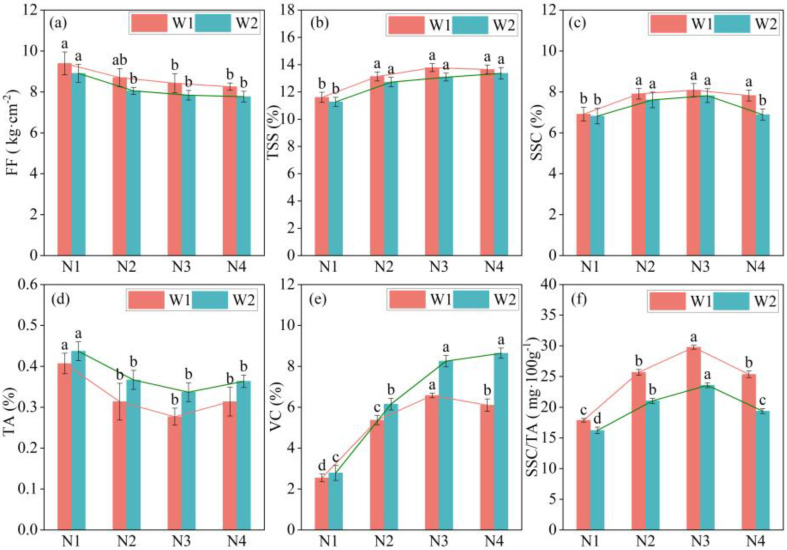
Response of intrinsic apple quality indices to water and nitrogen regulations in 2022. Histograms represent mean ± standard deviation (n = 3). Lower-case letters indicate that the mean values were significantly different at *p* ≤ 0.05. FF: flesh firmness; TSS: total soluble solids; SSC: soluble sugar content; TA: titratable acidity; VC: vitamin C; SSC/TA: sugar-acid ratio. Note: (**a**) is the FF content on apples in 2022, (**b**) is the TSS content on apples in 2022, (**c**) is the SSC content on apples in 2022, (**d**) is the TA content on apples in 2022, (**e**) is the VC content on apples in 2022, (**f**) is the SSC/TA content on apples in 2022.

**Figure 4 plants-13-02404-f004:**
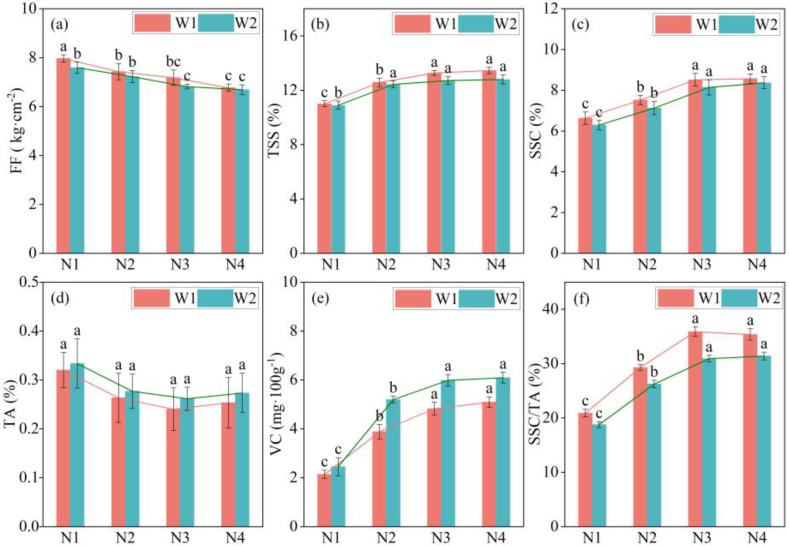
Response of apple intrinsic quality indices to water and nitrogen regulations in 2023. Histograms represent mean ± standard deviation (n = 3). Lower-case letters indicate that the mean values were significantly different at *p* ≤ 0.05. FF: flesh firmness; TSS: total soluble solids; SSC: soluble sugar content; TA: titratable acidity; VC: vitamin C; SSC/TA: sugar-acid ratio. Note: (**a**) is the FF content on apples in 2023, (**b**) is the TSS content on apples in 2023, (**c**) is the SSC content on apples in 2023, (**d**) is the TA content on apples in 2023, (**e**) is the VC content on apples in 2023, (**f**) is the SSC/TA content on apples in 2023.

**Figure 5 plants-13-02404-f005:**
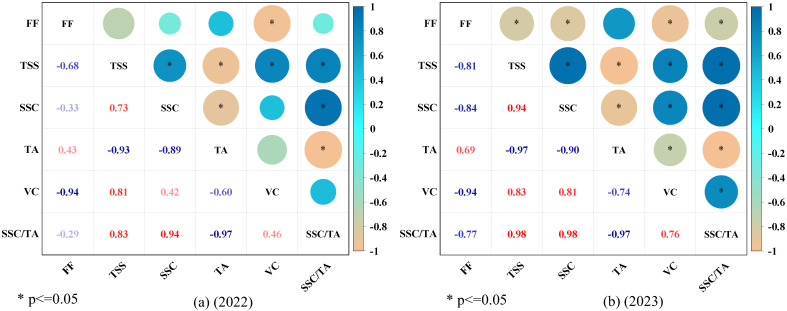
Pearson’s correlation coefficient between internal quality indices. * indicates significant correlation at the 0.05 level, respectively; sample size n = 24. FF: flesh firmness; TSS: total soluble solids; SSC: soluble sugar content; TA: titratable acidity; VC: vitamin C; SSC/TA: sugar-acid ratio. Note: The blue numbers in the figure indicate the negative correlation between the variables, and the magnitude of the value indicates the strength of the negative correlation. The red numbers indicate the positive correlation between the variables, and the magnitude of the value indicates the strength of the correlation.

**Figure 6 plants-13-02404-f006:**
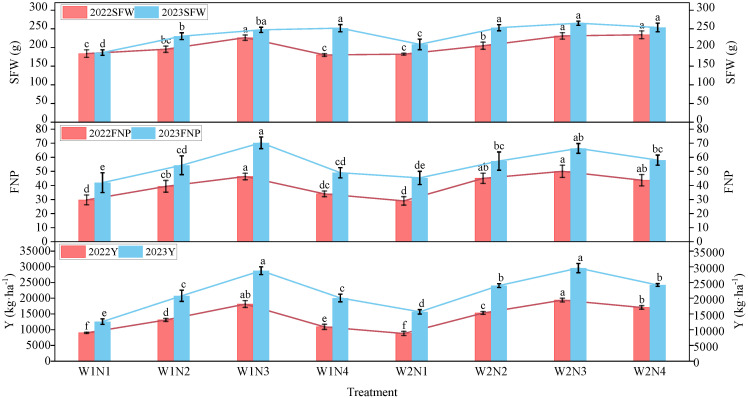
Responses of apple yield and components to water and nitrogen regulation. SFW, single-fruit weight; FNP, fruit number per plant; Y, yield. The same is below. Histograms represent mean ± standard deviation (n = 3). Lower-case letters indicate that the mean values were significantly different at *p* ≤ 0.05.

**Figure 7 plants-13-02404-f007:**
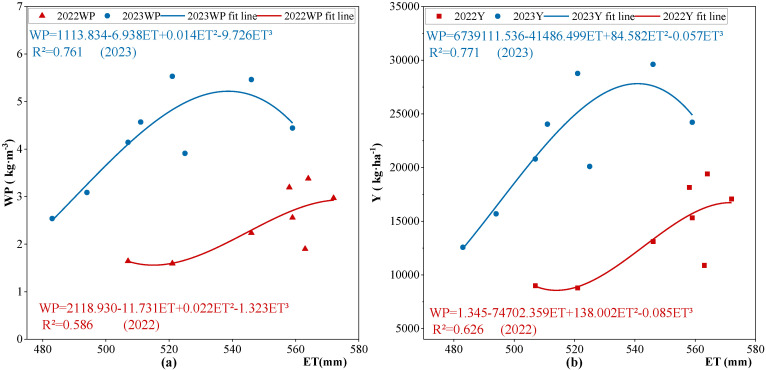
Effect of water consumption on water productivity and yield in apple trees. Scatter plots represent mean values (n = 3). WP: water productivity; Y: yield of apple trees; ET: water consumption of apple trees during phenological period. Note: (**a**) is relationship between ET and WP of apple trees, and (**b**) is the relationship between ET and Y of apple trees.

**Figure 8 plants-13-02404-f008:**
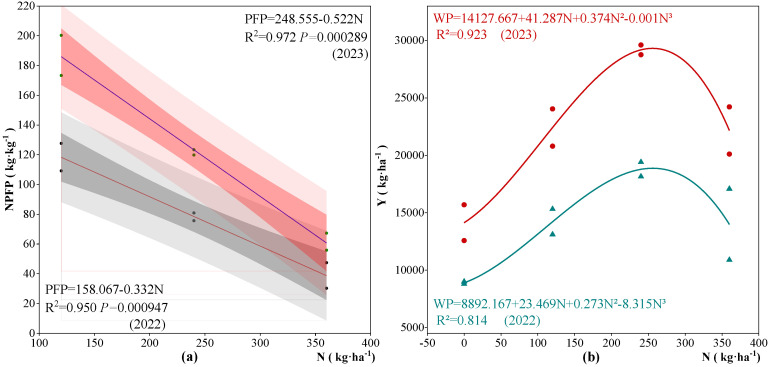
Effect of nitrogen application on apple tree nitrogen partial factor productivity and yield. Scatter plots represent mean values (n = 3). NPFP, nitrogen partial factor productivity; Y, yield of apple trees; N, nitrogen fertilizer application rate. Note: (**a**) is relationship between N and NPFP of apple trees, and (**b**) is the relationship between N and Y of apple trees.

**Figure 9 plants-13-02404-f009:**
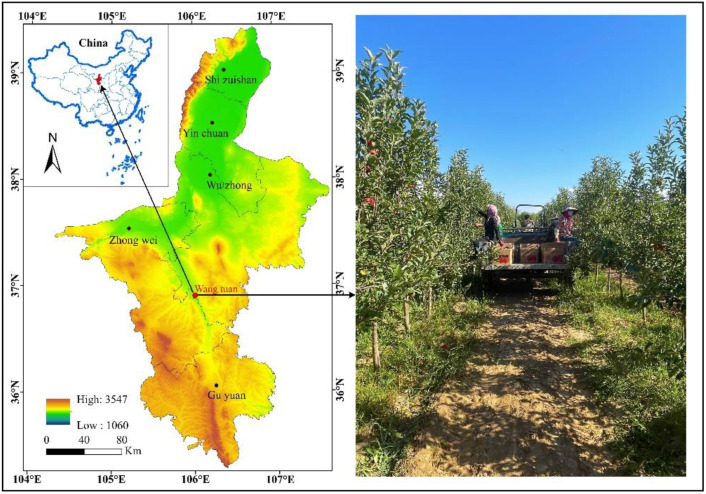
Location of the experimental site.

**Figure 10 plants-13-02404-f010:**
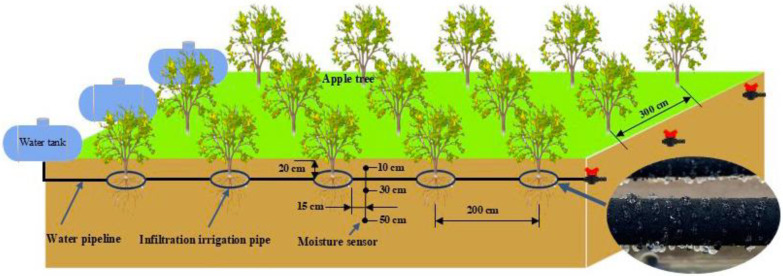
Layout of the subsurface infiltration irrigation system. The black circle in the figure represents underground seepage irrigation pipes arranged in an annular shape, with the tree trunk as the center (layout radius R = 30 cm).

**Figure 11 plants-13-02404-f011:**
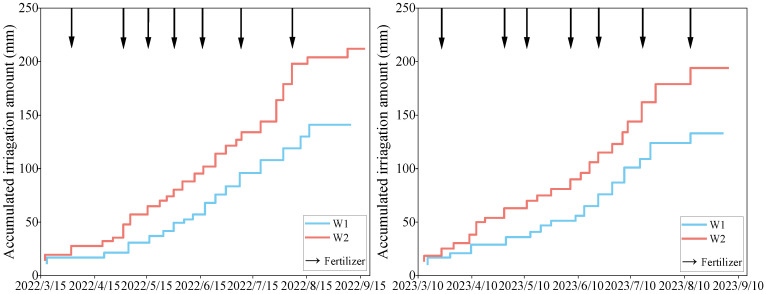
Cumulative irrigation amounts of the two water regulation treatments during the phenological period of apple trees.

**Figure 12 plants-13-02404-f012:**
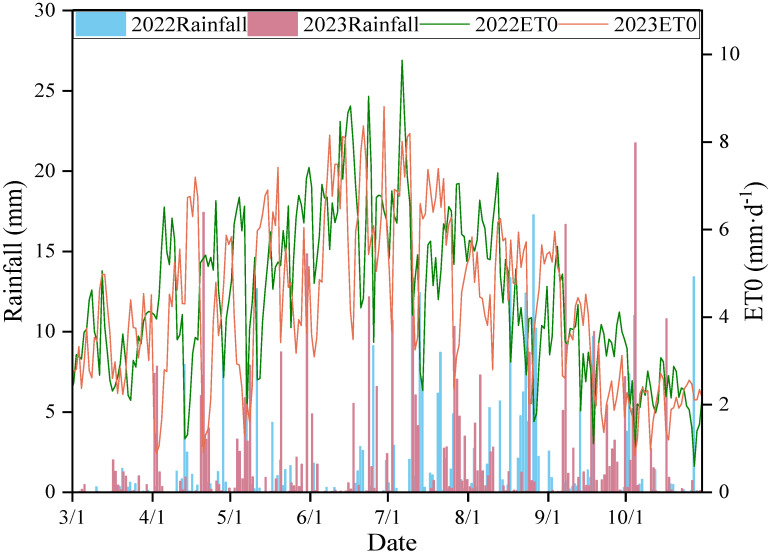
Rainfall and ET0 distribution of apple trees during two growing seasons (April–October) in 2022–2023.

**Table 1 plants-13-02404-t001:** ANOVA of the effect of water and nitrogen application amount on nitrogen content of apple tree leaves in 2022–2023.

Years	Factor	Flowing Stage	Perennial Growth Stage	Young Fruit Stage	Early Fruit Expansion	Rapid Fruit Expansion	Fruit Coloring Stage	Harvest Stage
2022	W	NS	NS	*	NS	NS	NS	NS
	N	NS	**	**	**	**	**	**
	W × N	*	NS	NS	NS	NS	NS	NS
2023	W	**	**	**	*	**	**	**
	N	**	**	**	**	**	**	**
	W × N	NS	NS	NS	NS	NS	NS	NS

Note: * and ** indicate significant influence at 0.05 and 0.01 levels, respectively, and NS indicates no significant influence.

**Table 2 plants-13-02404-t002:** Modeling and calculating the characteristic parameters of the spring growth process in different treatments.

Years	Treatments	Model Parameters			Characteristic Parameters			
		K (cm)	a	b	t_m_ (d)	v_m_ (cm·d^−1^)	T_m_ (d)	R^2^
2022	W1N1	27.63	2.21	1.51	20.93	0.58	69.59	0.928
	W1N2	22.81	3.03	4.18	10.60	1.11	25.23	0.990
	W1N3	20.42	3.67	4.81	10.81	1.24	21.90	0.993
	W1N4	20.37	2.88	3.96	10.70	1.05	26.63	0.991
	W2N1	19.41	2.97	4.07	10.72	1.07	25.91	0.993
	W2N2	23.65	2.85	4.10	10.22	1.08	25.72	0.991
	W2N3	21.95	3.46	4.35	11.41	1.13	24.20	0.992
	W2N4	21.62	2.86	3.66	11.47	0.98	28.78	0.994
2023	W1N1	18.74	2.22	3.17	10.05	0.90	33.25	0.994
	W1N2	30.08	4.12	2.72	20.78	0.96	38.68	0.994
	W1N3	27.45	3.97	3.56	15.50	1.04	29.59	0.999
	W1N4	26.94	3.50	3.18	15.78	0.98	33.17	0.997
	W2N1	20.29	3.39	2.88	16.97	0.90	36.62	0.998
	W2N2	31.21	4.76	2.37	26.32	0.95	44.46	0.997
	W2N3	30.59	4.41	2.71	21.91	0.97	38.88	0.997
	W2N4	28.31	4.76	2.86	21.82	1.01	36.85	0.999

Note: k is the maximum theoretical value of spring growth, t_m_ is the day with the highest growth rate of spring shoots, T_m_ is the day spring shoots stop growing, and v_m_ is the maximum spring growth rate.

**Table 3 plants-13-02404-t003:** Field experiment design for different water and nitrogen regulations in apple trees.

Treatment	The Lower Limit of Irrigation	Nitrogen Fertilizer Amount in Each Experiment Year (kg⸱ha^−1^)
W1N1	55%FC	0
W1N2	55%FC	120
W1N3	55%FC	240
W1N4	55%FC	360
W2N1	75%FC	0
W2N2	75%FC	120
W2N3	75%FC	240
W2N4	75%FC	360

Note: FC denotes the field capacity measured at field FC = 24.8% (volumetric).

## Data Availability

The data presented in this study are available within the article.

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
