# Peer review of "Effects of Water and Nitrogen Regulation on Apple Tree Growth, Yield, Quality, and Their Water and Nitrogen Utilization Efficiency"

_plants, 2024, doi:10.3390/plants13172404_

Round 1
Reviewer 1 Report
Comments and Suggestions for Authors
Language!!! It seems that authors used either translation services or AI. It is not a scientific language and all the text especially an abstract and introduction is overcrowded by fancy words and terms not used in scientific papers. Language must be revised!
Other reasons to reject the paper are:
Proper explanation is needed why two irrigation regimes were chosen, but control (optimal conditions) is missing!
Proper explanation is needed why such doses of N were chosen. Usually, 120 kg/ha is the highest recommended fertilisation rate.
Nitrogen and water regimes have effect not only on tree growth, yield and fruit quality in the current season, but also have impact on flower bud initiation, differentiation and the next year flowering and yield. It is not clear whether the same treatments were applied on the same trees during the 2 years of the experiment?
Some other additional remarks:
Use BBCH-scale instead or additionally to the description of the plant phenological stages as flowering period, new shoot growth period, young fruit period, early stage of fruit enlargement, rapid fruit enlargement stage, late stage of fruit enlargement, fruit coloring period etc.
There is a mixture of chapters. For example, 2.3.1. ‘Measurement of new shoot (spring) growth’ in Material and methods starts with a long introduction on shoot role in plant growth and nutrition. Such description here and in other places can be used in Discussion part, but not in MM. Be more precise!
Provide the information how many trees were in the replicate.
Comments on the Quality of English LanguageLook at the comments to authors
Author Response
Dear reviewer
We sincerely thank you for your professional comments on our article and your valuable comments and suggestions, which greatly improved the quality of our manuscript. Based on your comments and suggestions, we have made corresponding revisions to the manuscript. We have uploaded the revised manuscript and highlighted all the changes using the Track Changes mode in MS Word. We have also uploaded the manuscript with all the changes accepted. Attached to this letter, we have reproduced your revision comments and responded point by point. Our response is directly expressed in a different color (red) and the page and line number (blue) where the revision is located are marked. The line number in the response refers to the revised manuscript. The details are as follows.
Comment 1: Proper explanation is needed why two irrigation regimes were chosen, but control (optimal conditions) is missing!
Author's response: Thank you for the kind advice. In this study, the field water holding capacity of the 0-60 cm soil layer was used as the irrigation benchmark. With reference to local apple tree management technical regulations and related literature, two irrigation soil moisture conditions were set: 55% FC (severe drought) and 75% FC (moderate drought) for experiments.
Comment 2: Proper explanation is needed why such doses of N were chosen. Usually, 120 kg/ha is the highest recommended fertilisation rate.
Author's response: We sincerely thank you for your valuable suggestions. The nitrogen application gradient in the manuscript was set based on the actual soil conditions in the experimental area, and referred to the technical regulations for apple tree planting and management in Ningxia Hui Autonomous Region and related literature. For details, see the experimental design section (Section 2.2).
Comment 3: Nitrogen and water regimes have effect not only on tree growth, yield and fruit quality in the current season, but also have impact on flower bud initiation, differentiation and the next year flowering and yield. It is not clear whether the same treatments were applied on the same trees during the 2 years of the experiment?
Author's response: Thank you for the kind advice. This study was a positioning experiment, and the same treatment was applied to the same marked trees in both years of the experiment.
Comment 4: Use BBCH-scale instead or additionally to the description of the plant phenological stages as flowering period, new shoot growth period, young fruit period, early stage of fruit enlargement, rapid fruit enlargement stage, late stage of fruit enlargement, fruit coloring period etc.
Author's response: Thank you for the kind advice. The article has supplemented the apple tree growth phenological period recorded by the BBCH scale in the experimental design part. The author believes that the article focuses more on indicators such as leaf nitrogen content, canopy growth characteristics and yield quality in different growth stages of apples, rather than the morphological description of apple tree fruit growth.
Corrections: Note: According to the BBCH scale coding record, the main growth period of apple trees is divided into the budding stage (09); the new shoot growth stage (3) includes: the beginning of new shoot growth (31), about 30% of the final length (33), about 50% of the final length (35), about 70% of the final length (37), about 90% of the final length (39); flowering stage (65); fruit development stage (7) includes: fruit setting (71), early fruit enlargement stage (72), rapid fruit enlargement stage (74), late fruit enlargement stage (79); fruit maturity stage (8) includes: coloring stage (81) and maturity stage (89) [20,23]. (Modify the position: line 181-187, Page 5)
Comment 5: There is a mixture of chapters. For example, 2.3.1. ‘Measurement of new shoot (spring) growth’ in Material and methods starts with a long introduction on shoot role in plant growth and nutrition. Such description here and in other places can be used in Discussion part, but not in MM. Be more precise!
Author's response: Thank you for the kind advice. We have deleted the explanation part according to your suggestion and improved Section 2.3.1.
Corrections: Based on the actual growth conditions of the apple trees each year, observations of the spring shoot growth of the apple trees began at the initial stage of spring shoot growth. For each treatment, three apple trees with similar growth vigor were selected and marked. From each marked tree, three representative main branches were selected from the east, south, west, and north directions, and labeled sequentially. The length of the spring shoots on the main branches (excluding water sprouts) was measured with a steel ruler with an accuracy of 1 millimeter. The measurement period was approximately 10 days, and monitoring continued until the spring shoots stopped growing. (Modify the position: line 212-219, Page 6)
Comment 6: Provide the information how many trees were in the replicate.
Author's response: Thank you for the kind advice. In this study, three apple trees with similar growth characteristics were selected for marking in each treatment, with three replicates.
Corrections: a total of 8 treatment modalities (Table 1), three apple trees with similar growth characteristics were selected for marking in each treatment, each treatment was repeated three times to minimize plot effects. (Modify the position: line 153-156, Page 4)
Comment 7: Comments on the Quality of English Language — Look at the comments to authors
Author's response: Thank you very much for your valuable suggestions. We have invited an English teacher to help polish the language in the manuscript to make it more in line with standard academic expression. We sincerely thank you for your work again!

Reviewer 2 Report
Comments and Suggestions for Authors
Comments are in the attached file

Author Response
Dear reviewer
We sincerely thank you for your professional comments on our article and your valuable comments and suggestions, which greatly improved the quality of our manuscript. Based on your comments and suggestions, we have made corresponding revisions to the manuscript. We have uploaded the revised manuscript and highlighted all the changes using the Track Changes mode in MS Word. We have also uploaded the manuscript with all the changes accepted. Attached to this letter, we have reproduced your revision comments and responded point by point. Our response is directly expressed in a different color (red) and the page and line number (blue) where the revision is located are marked. The line number in the response refers to the revised manuscript. The details are as follows.
Comment 1: Keywords: can be supplemented with, for example, fruit quality characteristics
Author's response: Thank you for the kind advice. We added the keyword "fruit quality" based on your suggestion.
Corrections: Apple trees; Fruit quality; Logistic growth model; Subsurface infiltration irrigation; Water and nitrogen stress; Water and nitrogen supply decision-making (Modify the position: line 47-48, Page 2)
Comment 2: Introduction: Proposes to complete the chapter by defining the purpose of the study
Author's response: We sincerely thank you for your valuable suggestions. The last paragraph of the introduction of the manuscript describes the scientific question and purpose of this study and its importance.
Corrections: To summarise, research findings on the response of apple growth, development, yield, and quality to water and fertiliser availability are inconsistent. These discrepancies are related to the frequency and total amount of irrigation and nitrogen application, as well as the phenological stages of the fruit trees. Therefore, it is essential to systematically regulate water and fertiliser management techniques for high-quality and efficient apple production. Our work includes evaluating crop sensitivity to water and nitrogen during critical periods and determining yield and quality indicator reaction thresholds to soil moisture and nitrogen application. Furthermore, we used subsurface drip irrigation technology to supply water and nutrients to crops in a synchronous, even, timely, quantitative, and precise manner, resulting in the synchronous coupling of water and nutrients in time and space. This approach could improve the coordination and coupling effect of water and fertiliser supply in crop production, significantly increasing irrigation water and fertiliser utilisation efficiency. Our study can provide a scientific basis and guidelines for achieving high-quality and efficient apple production by controlling water and nitrogen levels. (Modify the position: line 113-127, Page 3)
Comment 3: Methods. It is necessary to complete the information on: area of experimental plots, number of apple trees, spacing, rootstock used, type of cultivation, tree treatments....
Author's response: Thank you for the kind advice. We've added relevant references based on your suggestions.
Corrections: The apple trees in the experimental area are "Huashuo" (an early-maturing variety; intermediate stock: M26 dwarfing rootstock; rootstock: Malus crabapple seedling rootstock). The tree rows are oriented north-south. The cultivation model is dense planting with dwarf rootstocks, with a planting distance of 200 cm × 300 cm and a planting density of 1600 trees per hectare. The trees have a free spindle shape, with an average height of approximately 290 cm. The average canopy width is 164 cm (north-south) × 157 cm (east-west). The fruit matures in mid to late August, with a relatively large size. The fruit can be stored at room temperature for about 30 days without the texture softening and can be stored under refrigeration for 3 months. It can be harvested early or fully ripened for concentrated harvesting [22]. (Modify the position: line 167-176, Page 4)
Comment 4: Chapter 2.3.1, lines 218-232. In my opinion, this is information that should not be included in the Methodology chapter. It has descriptive qualities and can be used in the discussion. What is missing, however, is practical information. What was the total number of shoots measured? Were measurements taken on the same shoots in both research seasons?
Author's response: Thank you for the kind advice. We have deleted the explanation part according to your suggestion and improved Section 2.3.1.
Corrections: Based on the actual growth conditions of the apple trees each year, observations of the spring shoot growth of the apple trees began at the initial stage of spring shoot growth. For each treatment, three apple trees with similar growth vigor were selected and marked. From each marked tree, three representative main branches were selected from the east, south, west, and north directions, and labeled sequentially. The length of the spring shoots on the main branches (excluding water sprouts) was measured with a steel ruler with an accuracy of 1 millimeter. The measurement period was approximately 10 days, and monitoring continued until the spring shoots stopped growing. (Modify the position: line 212-219, Page 6)
Comment 5: Line 154 What was the form of nitrogen - nitrate, ammonium?
Author's response: Thank you for the kind advice. The form of nitrogen mentioned in Line 154 of the article is alkaline hydrolyzed nitrogen. Alkaline hydrolyzed nitrogen is also called hydrolyzed nitrogen. It includes inorganic nitrogen and organic nitrogen with a simple structure that can be directly absorbed and utilized by crops. It can be absorbed and utilized by crops in the short term, so it is also called fast-acting nitrogen. It can reflect the nitrogen supply capacity of the soil in the short term.
Corrections: with alkaline hydrolyzed nitrogen of 43.46 mg·kg-1 (Modify the position: line 141, Page 3)
Comment 6: It is necessary to clarify water doses in irrigation in different years, which differ in the amount of precipitation.
Author's response: Thank you for the kind advice. Figure 3 shows the irrigation volumes for the two trial years 2022 and 2023.
Comment 7: Line 247. Inacurate description. Were 25 leaves taken from a tree or a combination? At how many dates were the leaves taken? In how many replicates were the analyses performed?
Author's response: Thank you very much for your suggestions and reminders. I am very sorry that my expression was inaccurate and caused difficulty in understanding. According to your suggestions, I revised the sentences in this section and described the sample preparation and test process in more detail.
Corrections: Leaf samples from the apple trees were collected during flowering, new shoot growth, young fruit, early fruit expansion, rapid fruit expansion, late fruit expansion, fruit coloring, and harvest stages. For each treatment, six current-year shoots were selected from the upper and lower parts and around the canopy of the marked apple trees. Four healthy, undamaged, mature leaves (without petioles) were collected from each shoot and placed in a single preservation bag. The labeled bags were placed in a cooler with ice packs and brought back to the laboratory. The leaves were washed with distilled water to remove surface dust and fuzz from the underside. The leaves were dried in a forced-air oven at 85°C for 30 minutes, then at 70°C until they reached a constant weight. Once cooled, the samples were ground into powder using a mortar and pestle. The ground leaves were passed through a 0.45 mm sieve and sealed for later use. A 0.2 g sample was placed into a digestion tube, and 5 mL of concentrated sulfuric acid (H2SO4) was added. The digestion tube was heated to 380°C, with repeated additions of hydrogen peroxide (H2O2) during heating until the solution became clear. After cooling, the digested solution was transferred to a 100 mL volumetric flask and diluted to volume. After mixing, 15 mL was taken with a pipette, and the nitrogen content of the leaves was determined using the Kjeldahl method (K1100F). (Modify the position: line 221-237, Page 7)
Comment 8: From how many trees were fruits harvested for measurements? The term "at least 10 fruits" is not precise. In how many replicates were the analyses performed?
Author's response: Thank you very much for your suggestion! We have revised Section 2.3.4 based on your suggestion and described in detail the storage method of the test samples and the experimental process of fruit quality determination.
Corrections: During the fruit maturity and harvest period, 3 representative fruits were selected from the east, west, south, and north directions of each tree, totaling 12 fruits per treatment. The harvested sample fruits were placed in preservation boxes and taken back to the laboratory, where they were stored in a refrigerator at 5°C. Quality analysis was completed within one week, employing a GY-4 digital hardness tester to gauge fruit flesh hardness; the Anthrone-sulfuric acid colorimetric method is used to ascertain the soluble sugar content in the fruits; a handheld refractometer is utilized to gauge the soluble solid content in the fruits; the vitamin C content in the fruits is determined using the molybdenum blue colorimetric method; and the titration method with sodium hydroxide is employed to measure the titratable acidity content in the fruits. (Modify the position: line 251-255, Page 7)
Comment 9: Line 326. Producent oprogramowania SPSS 22.0?
Author's response: Thank you for the kind advice. We have marked the corresponding positions in the article according to your suggestions.
Corrections: Analysis of variance (ANOVA) was performed using SPSS 22.0 software (IBM USA). (Modify the position: line 311, Page 9)
Comment 10: Discussion. The authors conclude that the highest leaf nitrogen content was found at the highest dose of nitrogen fertilizer. Such a result is obvious. The most important question - what do the authors think should be considered a stress factor? Maybe a very high dose of nitrogen - 360 kg ha-1?
Author's response: Thank you very much for your valuable suggestions. We have rephrased some sentences in the discussion section of the manuscript according to your suggestions. In this study, soil moisture and nitrogen fertilizer application rate were set as two factors, and the changes in fruit tree growth and physiological indicators under different irrigation gradients and nitrogen fertilizer application rates were monitored. Finally, all indicators were comprehensively evaluated to select the optimal irrigation limit and nitrogen fertilizer application rate based on the goal of improving or improving economic benefits.
Corrections: In our study, the nitrogen content in the leaves was highest under high nitrogen treatment (N4, 360 kg·ha-1), significantly higher than the no nitrogen treatment (N1). However, there was no significant difference compared to the other nitrogen treatments (N2, N3). This could be because the high nitrogen treatment increased the available nitrogen in the root zone beyond the tree's requirements; however, the tree could not fully assimilate the excess nitrogen [12]. (Modify the position: line 490-495, Page 17)
Comment 11: In the title of Chapters 4.1, 4.2 is the term "water stress." In the chapter content does not mention water stress. In the chapter authors make almost no reference to water stress. The question should be repeated - what do the authors mean by "water stress"? What was its measure? What effect does it have on the parameters studied? I refer the authors to the title of the article.
Author's response: We sincerely thank you for your valuable suggestions. Chapters 4.1 and 4.2 of the manuscript discuss the effects of different water and nitrogen supplies on leaf nitrogen content, shoot growth, yield and fruit quality of apple trees, where water stress refers to the two irrigation gradients in the experimental design, that is, different degrees of drought stress (the reference benchmark is the field water holding capacity of the experimental area), and the synergistic effect of water and nitrogen fertilizer is taken into account.

Reviewer 3 Report
Comments and Suggestions for Authors
Review on “Effects of water and nitrogen stress on growth, yield, and quality of apple trees and water and nitrogen utilization efficiency”.
The authors examined the effects of two irrigation thresholds and four nitrogen doses on one apple tree variety’s yield quality and quantity parameters, water and nitrogen use efficiency, during two growing seasons.
Understanding the interplay between water and nitrogen stress and their effects on apple growth, yield, and quality, as well as optimizing water and nitrogen utilization efficiency, is essential for sustainable orchard management and maximizing apple production under varying environmental conditions.
The research topic is very up-to-date work focusing on precision irrigation based on the use of different irrigation thresholds and reasonable fertilization management. However, not clear why the apple tree was selected as a test plant. The aim of the manuscript corresponds perfectly well with the scope of Plants-MPDI.
The manuscript is clear and well-organized. The manuscript contains many valuable results. Results are statistically analyzed. However, the manuscript needs some improvements and corrections.
Please restructure the manuscript and follow the Instruction for Authors guidelines.
https://www.mdpi.com/journal/plants/instructions
Abstract:
The abstract is well-written and concise. Properly explained the research aims and methods applied to achieve the objectives of the study.
Keywords:
Please arrange the keywords in alphabetical order.
Introduction:
The introduction is well-written and properly explains the justification of the study but mainly focuses on the importance of irrigation and the importance of adequate fertilization. Please add more information related to the importance of apple cultivation. Why was apple used as a test plant in this study? The authors please add more information about the situation of apple production in China or/and the importance of apple production in China.
Materials and methods:
Line 180-183: Please add a reference.
Line 218-240. Please revise this section. Explanations do not belong to the Materials and Methods section. Please delete all of the explanations. The materials and Methods section must only focus on the used materials and applied methods in the current study. If the authors want to use explanatory sentences please remove them from the Discussion section and use references as well.
Line 249: What was the purpose of „blanching”? Please explain it. At 105 C degree protein and nitrogen-containing substances are damaged so the measured weight is not reliable. I am sorry but the sample preparation method for the measurement of leaf nitrogen content does not seem accurate to me.
Line 262: Which kind of scale was used to measure the weight? Please be precise and add more information like analytical scale, brand of the appliance, country of origin, etc.
Line 267: How were the samples stored during the one week? Please add this information to the Materials and Methods section.
Results:
Table 4. Please use a horizontal line to separate the two growing seasons as in the case of Table 3. The quality of the figures and tables is fine.
Conclusions:
Some of the conclusions can't be accepted as new research findings like
- under the same nitrogen application level, increasing the irrigation threshold helps improve apple trees' leaf nitrogen content,
- nitrogen application significantly increases the nitrogen content of apple tree canopy leaves,
- nitrogen application significantly increases apple yield and optimizes yield components,
because many publications have already stated the same results using other test plants. Please rewrite these sentences to be more precise and focus on this study’s specifications.
Author Response
Dear reviewer
We sincerely thank you for your professional comments on our article and your valuable comments and suggestions, which greatly improved the quality of our manuscript. Based on your comments and suggestions, we have made corresponding revisions to the manuscript. We have uploaded the revised manuscript and highlighted all the changes using the Track Changes mode in MS Word. We have also uploaded the manuscript with all the changes accepted. Attached to this letter, we have reproduced your revision comments and responded point by point. Our response is directly expressed in a different color (red) and the page and line number (blue) where the revision is located are marked. The line number in the response refers to the revised manuscript. The details are as follows.
Comments and Suggestions for Authors
Review on “Effects of water and nitrogen stress on growth, yield, and quality of apple trees and water and nitrogen utilization efficiency”.
The authors examined the effects of two irrigation thresholds and four nitrogen doses on one apple tree variety’s yield quality and quantity parameters, water and nitrogen use efficiency, during two growing seasons.
Understanding the interplay between water and nitrogen stress and their effects on apple growth, yield, and quality, as well as optimizing water and nitrogen utilization efficiency, is essential for sustainable orchard management and maximizing apple production under varying environmental conditions.
The research topic is very up-to-date work focusing on precision irrigation based on the use of different irrigation thresholds and reasonable fertilization management. However, not clear why the apple tree was selected as a test plant. The aim of the manuscript corresponds perfectly well with the scope of Plants-MPDI.
The manuscript is clear and well-organized. The manuscript contains many valuable results. Results are statistically analyzed. However, the manuscript needs some improvements and corrections.
Authors’ response: We sincerely thank you for your positive and encouraging comments on our manuscript. We are honored to receive your recommendation. The following is a reply to your comments one by one. We sincerely hope that the answer to your question can satisfy you and gain your approval.
Comment 1: Abstract: The abstract is well-written and concise. Properly explained the research aims and methods applied to achieve the objectives of the study.
Author's response: Thank you very much for your affirmation of our manuscript!
Comment 2: Keywords: Please arrange the keywords in alphabetical order.
Author's response: Thank you for your suggestion. We have rearranged the keywords in alphabetical order according to your suggestion.
Corrections: Keywords: Apple trees; Logistic growth model; Subsurface infiltration irrigation; Water and nitrogen stress; Water and nitrogen supply decision-making (Modify the position: line 47-48, Page 2)
Comment 3: Introduction: The introduction is well-written and properly explains the justification of the study but mainly focuses on the importance of irrigation and the importance of adequate fertilization. Please add more information related to the importance of apple cultivation. Why was apple used as a test plant in this study? The authors please add more information about the situation of apple production in China or/and the importance of apple production in China.
Author's response: Thank you for the kind advice. We have rewritten this paragraph based on your suggestion that we consult the relevant information, describing China's apple production and acreage, as well as the share of economic value created by apple trees as a cash crop.
Corrections: Water resources are becoming increasingly scarce in many arid and semi-arid regions [1]. Irrigation for food production consumes most of the world’s freshwater, accounting for approximately 70% of the total water resources [2]. Crops, however, effectively use less than 60% of the irrigation water [3]. The proportion of economic forestry and fruit cultivation in agricultural planting systems has been increasing annually in tandem with social development. By 2023, the apple planting area would have reached 2.23×10^6 ha, with an annual fresh fruit production of 41.39×10^6 t, accounting for 45.2% of the world's planting area and 49.87% of global fresh fruit production [4]. Apple yields and biomass are higher than those of conventional annual crops, making water and nutrients critical for efficient and high-quality apple production. A lack of scientific guidance, traditional farmer beliefs, and the combined impact of soil and production environments result in inefficient water and fertiliser usage in orchards. This leads to economic losses for fruit farmers and several environmental challenges, including nitrogen leaching and volatilisation [5]. Considering the importance of water-nitrogen interactions, water and nitrogen management in fruit tree production processes requires further in-depth research. (Modify the position: line 51-66, Page 2)
Comment 4: Materials and methods
Comment 4.1: Line 180-183: Please add a reference.
Author's response: Thank you for the kind advice. We've added relevant references based on your suggestions.
Corrections: The apple trees in the experimental area are "Huashuo" (an early-maturing variety; intermediate stock: M26 dwarfing rootstock; rootstock: Malus crabapple seedling rootstock). The tree rows are oriented north-south. The cultivation model is dense planting with dwarf rootstocks, with a planting distance of 200 cm × 300 cm and a planting density of 1600 trees per hectare. The trees have a free spindle shape, with an average height of approximately 290 cm. The average canopy width is 164 cm (north-south) × 157 cm (east-west). The fruit matures in mid to late August, with a relatively large size. The fruit can be stored at room temperature for about 30 days without the texture softening and can be stored under refrigeration for 3 months. It can be harvested early or fully ripened for concentrated harvesting [22]. (Modify the position: line 167-176, Page 4)
- Yan, Z.; Zhang, H.; Guo, G.; Zhang, S.; Liu, Z. Selection of a new early-ripening apple cultivar———Huashuo. Journal of Fruit Science 2010, 27, 655-656+480, doi:10.13925/j.cnki.gsxb.2010.04.033.
Comment 4.2: Line 218-240. Please revise this section. Explanations do not belong to the Materials and Methods section. Please delete all of the explanations. The materials and Methods section must only focus on the used materials and applied methods in the current study. If the authors want to use explanatory sentences please remove them from the Discussion section and use references as well.
Author's response: Thank you for the kind advice. We have deleted the explanation part according to your suggestion and improved Section 2.3.1.
Corrections: Based on the actual growth conditions of the apple trees each year, observations of the spring shoot growth of the apple trees began at the initial stage of spring shoot growth. For each treatment, three apple trees with similar growth vigor were selected and marked. From each marked tree, three representative main branches were selected from the east, south, west, and north directions, and labeled sequentially. The length of the spring shoots on the main branches (excluding water sprouts) was measured with a steel ruler with an accuracy of 1 millimeter. The measurement period was approximately 10 days, and monitoring continued until the spring shoots stopped growing. (Modify the position: line 212-219, Page 6)
Comment 4.3: Line 249: What was the purpose of “blanching”? Please explain it. At 105 C degree protein and nitrogen-containing substances are damaged so the measured weight is not reliable. I am sorry but the sample preparation method for the measurement of leaf nitrogen content does not seem accurate to me.
Author's response: Thank you very much for your suggestions and reminders. I am very sorry that my expression was inaccurate and caused difficulty in understanding. According to your suggestions, I revised the sentences in this section and described the sample preparation and test process in more detail.
Corrections: Leaf samples from the apple trees were collected during flowering, new shoot growth, young fruit, early fruit expansion, rapid fruit expansion, late fruit expansion, fruit coloring, and harvest stages. For each treatment, six current-year shoots were selected from the upper and lower parts and around the canopy of the marked apple trees. Four healthy, undamaged, mature leaves (without petioles) were collected from each shoot and placed in a single preservation bag. The labeled bags were placed in a cooler with ice packs and brought back to the laboratory[25]. The leaves were washed with distilled water to remove surface dust and fuzz from the underside. The leaves were dried in a forced-air oven at 85°C for 30 minutes, then at 70°C until they reached a constant weight. Once cooled, the samples were ground into powder using a mortar and pestle. The ground leaves were passed through a 0.45 mm sieve and sealed for later use. A 0.2 g sample was placed into a digestion tube, and 5 mL of concentrated sulfuric acid (H2SO4) was added. The digestion tube was heated to 380°C, with repeated additions of hydrogen peroxide (H2O2) during heating until the solution became clear. After cooling, the digested solution was transferred to a 100 mL volumetric flask and diluted to volume. After mixing, 15 mL was taken with a pipette, and the nitrogen content of the leaves was determined using the Kjeldahl method (K1100F) [26] (Modify the position: line 221-237, Page 6 and 7)
Comment 4.4: Line 262: Which kind of scale was used to measure the weight? Please be precise and add more information like analytical scale, brand of the appliance, country of origin, etc.
Author's response: Thank you for the kind advice. I have improved and modified the apple yield determination process based on your suggestions.
Corrections: During the fruit maturity and harvest period, the field weighing method was used to harvest the marked apple trees according to different treatments. The yield per tree was measured with an electronic balance (0.01 g) (Precisa XJ3200C, China), and the number of fruits per tree and the weight of individual fruits were recorded. Then, the total yield per hectare for each treatment was calculated based on the planting density. (Modify the position: line 245-249, Page 7)
Comment 4.5: Line 267: How were the samples stored during the one week? Please add this information to the Materials and Methods section.
Author's response: Thank you very much for your suggestion! We have revised Section 2.3.4 based on your suggestion and described in detail the storage method of the test samples and the experimental process of fruit quality determination.
Corrections: During the fruit maturity and harvest period, 3 representative fruits were selected from the east, west, south, and north directions of each tree, totaling 12 fruits per treatment. The harvested sample fruits were placed in preservation boxes and taken back to the laboratory, where they were stored in a refrigerator at 5°C. Quality analysis was completed within one week, employing a GY-4 digital hardness tester to gauge fruit flesh hardness; the Anthrone-sulfuric acid colorimetric method is used to ascertain the soluble sugar content in the fruits; a handheld refractometer is utilized to gauge the soluble solid content in the fruits; the vitamin C content in the fruits is determined using the molybdenum blue colorimetric method; and the titration method with sodium hydroxide is employed to measure the titratable acidity content in the fruits [27]. (Modify the position: line 251-260, Page 7)
Comment 5: Results: Table 4. Please use a horizontal line to separate the two growing seasons as in the case of Table 3. The quality of the figures and tables is fine.
Author's response: Thank you very much for your suggestions and reminders! We have modified Table 4 according to your suggestions.
Corrections: (Modify the position: line 380, Page 12)
|
Years |
Treatments |
Model parameters |
Characteristic parameters |
|||||
|
k(cm) |
a |
b |
tm (d) |
vm(cm·d-1) |
Tm(d) |
R2 |
||
|
2022 |
W1N1 |
27.63 |
2.21 |
1.51 |
20.93 |
0.58 |
69.59 |
0.928 |
|
W1N2 |
22.81 |
3.03 |
4.18 |
10.60 |
1.11 |
25.23 |
0.990 |
|
|
W1N3 |
20.42 |
3.67 |
4.81 |
10.81 |
1.24 |
21.90 |
0.993 |
|
|
W1N4 |
20.37 |
2.88 |
3.96 |
10.70 |
1.05 |
26.63 |
0.991 |
|
|
W2N1 |
19.41 |
2.97 |
4.07 |
10.72 |
1.07 |
25.91 |
0.993 |
|
|
W2N2 |
23.65 |
2.85 |
4.10 |
10.22 |
1.08 |
25.72 |
0.991 |
|
|
W2N3 |
21.95 |
3.46 |
4.35 |
11.41 |
1.13 |
24.20 |
0.992 |
|
|
W2N4 |
21.62 |
2.86 |
3.66 |
11.47 |
0.98 |
28.78 |
0.994 |
|
|
2023 |
W1N1 |
18.74 |
2.22 |
3.17 |
10.05 |
0.90 |
33.25 |
0.994 |
|
W1N2 |
30.08 |
4.12 |
2.72 |
20.78 |
0.96 |
38.68 |
0.994 |
|
|
W1N3 |
27.45 |
3.97 |
3.56 |
15.50 |
1.04 |
29.59 |
0.999 |
|
|
W1N4 |
26.94 |
3.50 |
3.18 |
15.78 |
0.98 |
33.17 |
0.997 |
|
|
W2N1 |
20.29 |
3.39 |
2.88 |
16.97 |
0.90 |
36.62 |
0.998 |
|
|
W2N2 |
31.21 |
4.76 |
2.37 |
26.32 |
0.95 |
44.46 |
0.997 |
|
|
W2N3 |
30.59 |
4.41 |
2.71 |
21.91 |
0.97 |
38.88 |
0.997 |
|
|
W2N4 |
28.31 |
4.76 |
2.86 |
21.82 |
1.01 |
36.85 |
0.999 |
|
Comment 6: Conclusions: Some of the conclusions can't be accepted as new research findings like
-under the same nitrogen application level, increasing the irrigation threshold helps improve apple trees' leaf nitrogen content,
-nitrogen application significantly increases the nitrogen content of apple tree canopy leaves,
-nitrogen application significantly increases apple yield and optimizes yield components,
because many publications have already stated the same results using other test plants. Please rewrite these sentences to be more precise and focus on this study’s specifications.
Author's response: Thank you very much for your suggestion. We have rewritten some sentences in the conclusion chapter to make it more accurately consistent with the conclusions of this study.
Corrections: This study investigated the response of apple trees to various water and nitrogen stress conditions in terms of nitrogen nutrition, canopy growth characteristics, yield quality, and water-nitrogen use efficiency. At the same nitrogen application level, the W2 (75% FC) treatment had higher leaf nitrogen content than the W1 (55% FC) treatment, although the difference was not significant (P>0.05). Nitrogen administration significantly improved leaf nitrogen content compared to the no nitrogen (N1) treatment (P<0.05). The growth pattern of new shoots was consistent with the logistic growth curve. High water and high nitrogen treatments could delay the maximum new shoot growth rate and extend the growth period of the new shoots. Apple tree yields at the N3 level were highest at 19405.3 kg·ha-1 (2022) and 29607 kg·ha-1 (2023). Apple yield was found to have a parabolic relationship with nitrogen application rate, with 230-260 kg·ha-1 being the theoretically optimal range for nitrogen application. The nitrogen application rate significantly affected the quality indices of apples, but the lower irrigation limit had no significant effect (P>0.05). At the same nitrogen application level, the W2 treatment improved the utilisation efficiency of water and nitrogen fertiliser. The amount of nitrogen fertiliser negatively affected the partial productivity of nitrogen fertiliser. The W2N2 treatment resulted in agronomic nitrogen fertiliser efficiency of 127.6 kg·kg-1 (2022) and 200.3 kg·kg-1 (2023), while the W2N3 treatment improved fruit quality and water productivity significantly while maintaining apple yield. (Modify the position: line 582-600, Page 19)

Reviewer 4 Report
Comments and Suggestions for Authors
The article evaluates the effect of two watering regimes and four nitrogen fertilization plans in apple trees cv ‘Huashuo’. Trials were conducted on an open-field apple orchard located in Ningxia Region, China, and encompassed two productive seasons, namely years 2022 and 2023. Analyses focused on both phenological and productive traits, but also nitrogen and water use efficiency. For each year, researchers compared the overall performance ascribed to each irrigation x fertilizer combination and were able to consistently detect the most appropriate combination between these two factors. For this reason, the present study represents a remarkable contribution to horticultural sciences and stakeholders operating in this sector.
2. General concept comments
2.1. Article
The article has a solid and sound presentation; materials and methods are well organized and carefully explained. Analyses are well-rounded and investigate the topic from many points of view, giving the reader lots of information to speculate upon. Relying on two years instead of just one gives this work even more robustness from a scientific point of view.
In my opinion, both the abstract and the introduction should go through a change of tone in terms of English vocabulary: the information is there, and it is satisfactory, but sentences are arranged in an excessively formal way, thus hampering the delivery of the main message. Even the choice of the words is, at times, a bit too much “aristocratic” and not so compliant with the vocabulary that should be used within a research paper.
2.2. Review
Overall, the present study represents a solid piece of research which might spark more insights and knowledge among the scientific community. The elaborations are clear and original and features many different approaches through which data were processed.
Moreover, it also holds very practical outcomes which can turn out to be exceptionally useful for apple growers and other stakeholders having a role within the apple cultivation sector.
1. Specific comments
Line 25: “sans” is not an English word, to my knowledge; in any case, I suggest using a more common word such as “without”;
Line 31: The N3 treatment is mentioned without it being explained beforehand; I recommend explaining what each code refers to before adopting it within the body text;
Line 168: In my opinion, “Reigned supreme” does not really comply with the style of a scientific article;
Line 181: I could not understand what “without turning to sand” actually meant. I would suggest writing this sentence over in a clearer way;
Figure 6: the scale used for the spring shoot length is not identical among the four graphs (goes up to 26cm on the 2023W1 graph, while reaching 24cm in all the others). I would suggest harmonizing the scale;
Line 399: I would not include titratable acidity among the “positive indicators” since it does not belong to this category, as later explained by the authors themselves;
Figure 11: I suggest writing a clearer caption for this figure as it does not clearly explain what each colour and line refer to; moreover, the graph concerning WP seems to feature a 2023 and a 2022 line, while the Y graph shows two 2023 lines; in case there was a typo in any of the two graphs composing fig. 11, I would suggest correcting it.
Comments on the Quality of English LanguageMinor editing of English language required especially in abstract and introduction sections.
Author Response
Dear reviewer
We sincerely thank you for your professional comments on our article and your valuable comments and suggestions, which greatly improved the quality of our manuscript. Based on your comments and suggestions, we have made corresponding revisions to the manuscript. We have uploaded the revised manuscript and highlighted all the changes using the Track Changes mode in MS Word. We have also uploaded the manuscript with all the changes accepted. Attached to this letter, we have reproduced your revision comments and responded point by point. Our response is directly expressed in a different color (red) and the page and line number (blue) where the revision is located are marked. The line number in the response refers to the revised manuscript. The details are as follows.
- Comments and Suggestions for Authors
The article evaluates the effect of two watering regimes and four nitrogen fertilization plans in apple trees cv ‘Huashuo’. Trials were conducted on an open-field apple orchard located in Ningxia Region, China, and encompassed two productive seasons, namely years 2022 and 2023. Analyses focused on both phenological and productive traits, but also nitrogen and water use efficiency. For each year, researchers compared the overall performance ascribed to each irrigation x fertilizer combination and were able to consistently detect the most appropriate combination between these two factors. For this reason, the present study represents a remarkable contribution to horticultural sciences and stakeholders operating in this sector.
- General concept comments
2.1. Article
The article has a solid and sound presentation; materials and methods are well organized and carefully explained. Analyses are well-rounded and investigate the topic from many points of view, giving the reader lots of information to speculate upon. Relying on two years instead of just one gives this work even more robustness from a scientific point of view.
In my opinion, both the abstract and the introduction should go through a change of tone in terms of English vocabulary: the information is there, and it is satisfactory, but sentences are arranged in an excessively formal way, thus hampering the delivery of the main message. Even the choice of the words is, at times, a bit too much “aristocratic” and not so compliant with the vocabulary that should be used within a research paper.
2.2. Review
Overall, the present study represents a solid piece of research which might spark more insights and knowledge among the scientific community. The elaborations are clear and original and features many different approaches through which data were processed.
Moreover, it also holds very practical outcomes which can turn out to be exceptionally useful for apple growers and other stakeholders having a role within the apple cultivation sector.
Authors’ response: We sincerely thank you for your positive and encouraging comments on our manuscript. We are honored to receive your recommendation. The following is a reply to your comments one by one. We sincerely hope that the answer to your question can satisfy you and gain your approval.
Specific comments
Comment 1: Line 25: “sans” is not an English word, to my knowledge; in any case, I suggest using a more common word such as “without”;
Author's response: Thank you for the kind advice. We have revised the sentence based on your suggestion.
Corrections: The results showed that increasing the irrigation lower limit while maintaining the nitrogen application level enhanced the leaf nitrogen content of apple trees, but insignificantly (P>0.05). (Modify the position: line 27-30, Page 1)
Comment 2: Line 31: The N3 treatment is mentioned without it being explained beforehand; I recommend explaining what each code refers to before adopting it within the body text;
Author's response: Thank you for the kind advice. According to your suggestion, we have explained the nitrogen treatment in the experiment so that readers can understand it more clearly.
Corrections: A randomized block design was employed with two irrigation lower limits (W1(55% FC), W2(75% FC)) and four nitrogen application levels (N1(0 kg·ha-2), N2(120 kg·ha-2), N3(240 kg·ha-2), N4(360 kg·ha-2)). (Modify the position: line 21-24, Page 1)
Comment 3: Line 168: In my opinion, “Reigned supreme” does not really comply with the style of a scientific article;
Author's response: Thank you for the kind advice. We have revised the sentence based on your suggestion.
Corrections: All treatments used the same amounts of phosphorus fertilizer (P2O5) and potash fertilizer (K2O) (225 kg⸱ha-1 for P2O5 and 300 kg⸱ha-1 for K2O in 2021-2022; 195 kg⸱ha-1 for P2O5 and 240 kg⸱ha-1 for K2O in 2022-2023). (Modify the position: line 156-159, Page 4)
Comment 4: Line 181: I could not understand what “without turning to sand” actually meant. I would suggest writing this sentence over in a clearer way;
Author's response: “without turning to sand” means the texture of the apple fruit softened. I am really sorry, maybe I did not use professional vocabulary, which led to misunderstanding. I have rewritten it according to your suggestion. Thank you very much for your reminder!
Corrections: The fruit can be stored at room temperature for about 30 days without the texture softening and can be stored under refrigeration for 3 months. It can be harvested early or fully ripened for concentrated harvesting [22]. (Modify the position: line 173-176, Page 4)
Comment 5: Figure 6: the scale used for the spring shoot length is not identical among the four graphs (goes up to 26cm on the 2023W1 graph, while reaching 24cm in all the others). I would suggest harmonizing the scale;
Author's response: Thank you for the kind advice. We have rescaled Figure 6 based on your suggestion.
Corrections: (Modify the position: line 377, Page 12)
Comment 6: Line 399: I would not include titratable acidity among the “positive indicators” since it does not belong to this category, as later explained by the authors themselves;
Author's response: Thank you very much for your advice and reminder! I rechecked the wording of this sentence and made corrections.
Corrections: There are many intrinsic quality indicators for apples and differences in the directionalities of different indicators. For example, flesh firmness (FF), total soluble solids (TSS), soluble sugars content (SSC), vitamin C (VC), and sugar-acid ratio (SSC/TA) are all positive indicators, meaning that the larger the value, the better; In contrast, titratable acidity (TA) is a negative indicator, meaning that the smaller the value, the better. (Modify the position: line 386-390, Page 13)
Comment 7: Figure 11: I suggest writing a clearer caption for this figure as it does not clearly explain what each colour and line refer to; moreover, the graph concerning WP seems to feature a 2023 and a 2022 line, while the Y graph shows two 2023 lines; in case there was a typo in any of the two graphs composing fig. 11, I would suggest correcting it.
Author's response: Thank you for your reminder and suggestion. We are very sorry for our careless mistakes, and based on your suggestions, I have added a legend explaining the meaning of each line in the diagram to make it clearer and corrected the writing errors in the diagram.
Corrections: (Modify the position: line 457, Page 16)
Comment 8: Comments on the Quality of English Language —— Minor editing of English language required especially in abstract and introduction sections.
Author's response: Thank you very much for your valuable suggestions. We have invited an English teacher to help polish the language in the manuscript to make it more in line with standard academic expression. We sincerely thank you for your work again!

Round 2
Reviewer 1 Report
Comments and Suggestions for Authors
There are still left parts with not corrected language.
‘The locale boasts an annual mean temperature of 8.6°C and annual mean precipitation of 270 mm, characterized by an uneven seasonal apportionment of rainfall, predominantly clustered betwixt July and September. The multiannual evaporation quotient in the experimental ambit tallies 2325 mm, adorned with a frost-free interval oscillating betwixt 120 to 218 daily revolutions, alongside a multiannual solar diurnal illumination sumptuousness reaching 3024 hours.’
‘Four tiers of nitrogen application were outlined:’
‘Urea (N 46%) was the chosen nitrogen fertilizer, while phosphorus and potassium fertilizers assumed the guise of potassium dihydrogen phosphate (P2O5 52%, K2O 34%), with supplemental infusions of potassium chloride (K2O 60%) as deemed necessary’
‘reaching its nadir’; ‘a modest resurgence’, ‘impoverished soil’, ‘Nitrogen administration’ and so on.
Other remarks:
- All tables and figures should be understandable without referring to the text. In footnotes provide explanations of all abbreviations used.
- Both water regimes were deficit regimes. In conclusions you are mentioning high water treatment. What it is?
- Statement ‘The cultivation model is dense planting with dwarf rootstocks, with a planting distance of 200 cm × 300 cm and a planting density of 1600 trees per hectare’ is not correct. 2 m distance between trees, and semi dwarfing M.26 used as interstock indicate that it is not a ‘dense planting’.
- Statement ‘From each marked tree, three representative main branches were selected from the east, south, west, and north directions, and labeled sequentially’. How is it possible to select 3 branches from 4 directions? Or was it 12 branches selected?
Comments on the Quality of English Languageremarks provided in Comments for authors.
Author Response
Dear reviewer
We sincerely thank you for your professional comments on our article and your valuable comments and suggestions, which greatly improved the quality of our manuscript. Based on your comments and suggestions, we have made corresponding revisions to the manuscript. We have uploaded the revised manuscript and highlighted all the changes using the Track Changes mode in MS Word. We have also uploaded the manuscript with all the changes accepted. Attached to this letter, we have reproduced your revision comments and responded point by point. Our response is directly expressed in a different color (red) and the page and line number (blue) where the revision is located are marked. The line number in the response refers to the revised manuscript. The details are as follows.
There are still left parts with not corrected language.
‘The locale boasts an annual mean temperature of 8.6°C and annual mean precipitation of 270 mm, characterized by an uneven seasonal apportionment of rainfall, predominantly clustered betwixt July and September. The multiannual evaporation quotient in the experimental ambit tallies 2325 mm, adorned with a frost-free interval oscillating betwixt 120 to 218 daily revolutions, alongside a multiannual solar diurnal illumination sumptuousness reaching 3024 hours.’
‘Four tiers of nitrogen application were outlined:’
‘Urea (N 46%) was the chosen nitrogen fertilizer, while phosphorus and potassium fertilizers assumed the guise of potassium dihydrogen phosphate (P2O5 52%, K2O 34%), with supplemental infusions of potassium chloride (K2O 60%) as deemed necessary’
‘reaching its nadir’; ‘a modest resurgence’, ‘impoverished soil’, ‘Nitrogen administration’ and so on.
Author's response: We sincerely appreciate your comments. We have reviewed and revised the language in the article to make it more consistent with standard academic expression. We sincerely thank you for your work again!
Comment 1: All tables and figures should be understandable without referring to the text. In footnotes provide explanations of all abbreviations used.
Author's response: Thank you very much for your comments. According to your suggestion, we have revised the figures and tables in the manuscript without adding abbreviation annotations to help readers understand the contents of the figures more easily.
Comment 2: Both water regimes were deficit regimes. In conclusions you are mentioning high water treatment. What it is?
Author's response: We sincerely thank you for your comments. The ‘high water treatment’ mentioned in the conclusion of the manuscript refers to the treatment of the high irrigation limit (W2) in this study.
Comment 3: Statement ‘The cultivation model is dense planting with dwarf rootstocks, with a planting distance of 200 cm × 300 cm and a planting density of 1600 trees per hectare’ is not correct. 2 m distance between trees, and semi dwarfing M.26 used as interstock indicate that it is not a ‘dense planting’.
Author's response: Thank you for the kind advice. We have reviewed relevant materials based on your suggestions and made corrections in the manuscript.
Corrections: The tree rows are oriented north-south direction, with a planting distance of 200 cm × 300 cm and a planting density of 1600 trees per hectare. (Modify the position: line 444-445, Page 16)
Comment 3: Statement ‘From each marked tree, three representative main branches were selected from the east, south, west, and north directions, and labeled sequentially’. How is it possible to select 3 branches from 4 directions? Or was it 12 branches selected?
Author's response: Thank you for the kind advice. The statement in the manuscript that ‘From each marked tree, three representative main branches were selected from the east, south, west, and north directions, and labeled sequentially’ means that three branches were selected from each of the four directions (east, south, west and north) of the apple tree body for marking, for a total of 12 branches. We have rechecked and corrected the language in the manuscript based on your suggestion.
Corrections: Three apple trees with similar growth were selected for each treatment and marked. Then, three representative main branches were selected from the east, south, west and north directions of each marked apple tree, for a total of 12 branches. (Modify the position: line 487-490, Page 16)

Reviewer 2 Report
Comments and Suggestions for Authors
I thank the authors for the corrections and clarifications wchich have been made and wish good luck in their future scientific activities
Author Response
Sincerely thank you for your encouragement. We are honored to receive your recommendation. We will continue to work hard. Thank you again! I wish you good health and success in your work.
Reviewer 3 Report
Comments and Suggestions for Authors
Thank you for the corrections. However, one suggestion was overlooked, please check the Instruction for Authors https://www.mdpi.com/journal/plants/instructions
and restructure the manuscript. The order of the sections should be the following: Introduction, Results, Discussion, Materials and Methods, and Conclusions.
Please refer to your presented research data in the Discussion section using Table 1, Table 2, etc. when the authors compare their results with other scientists' results.
Author Response
We sincerely thank you for your positive and encouraging comments on our manuscript. We are honored to receive your recommendation and approval. We read the author guidelines according to your suggestions and adjusted the order of chapters as needed. We thank you again for your hard work in reviewing our manuscript and wish you good health and good work.
Round 3
Reviewer 1 Report
Comments and Suggestions for Authors
Thank you for the consideration of my remarks.
Author Response

(The authors gave the same response as above.)
